# Dynein-mediated transport and membrane trafficking control PAR3 polarised distribution

**Julie Jouette, Antoine Guichet\*, Sandra B Claret\***

Institut Jacques Monod, CNRS, UMR 7592, Paris Diderot University, Sorbonne Paris Cité, Paris, France

**Abstract** The scaffold protein PAR3 and the kinase PAR1 are essential proteins that control cell polarity. Their precise opposite localisations define plasma membrane domains with specific functions. PAR3 and PAR1 are mutually inhibited by direct or indirect phosphorylations, but their fates once phosphorylated are poorly known. Through precise spatiotemporal quantification of PAR3 localisation in the *Drosophila* oocyte, we identify several mechanisms responsible for its anterior cortex accumulation and its posterior exclusion. We show that PAR3 posterior plasma membrane exclusion depends on PAR1 and an endocytic mechanism relying on RAB5 and PI(4,5)P2. In a second phase, microtubules and the dynein motor, in connection with vesicular trafficking involving RAB11 and IKK-related kinase, IKKε, are required for PAR3 transport towards the anterior cortex. Altogether, our results point to a connection between membrane trafficking and dynein-mediated transport to sustain PAR3 asymmetry.

DOI: https://doi.org/10.7554/eLife.40212.001

**\*For correspondence:**
antoine.guichet@ijm.fr (AG);
sandra.claret@ijm.fr (SBC)

**Competing interests:** The authors declare that no competing interests exist.

## Introduction

Cell polarity is a fundamental process involved in diverse processes crucial for cell function and development. Establishment and maintenance of cell polarity are under the control of a set of polarity proteins highly conserved through metazoans. This network of proteins, some of which are PAR proteins, is set up through regulations involving recruitment or repulsion(*Assémat et al., 2008*; *Goldstein and Macara, 2007*; *Laprise and Tepass, 2011*; *Nelson, 2003*; *Rodriguez-Boulan and Macara, 2014*; *St Johnston and Ahringer, 2010*). The precise localisation of these polarity complexes defines plasma membrane domains with specific functions (*Rodriguez-Boulan and Macara, 2014*; *St Johnston and Ahringer, 2010*). Accordingly, the polarity complexes are essential for establishment of neural growth cone polarity (*Yu et al., 2006*), apico-basal epithelial polarity (*Kuchinke et al., 1998*; *Nance et al., 2003*), or antero-posterior polarity in *Drosophila* oocytes (*Cox et al., 2001b*; *Tomancak et al., 2000*) and *C. elegans* embryos (*Kemphues, 2000*).

Two major polarity modules control establishment and maintenance of cell polarity: one module composed of PAR3 (also named Bazooka [BAZ] in *Drosophila*), PAR6, and aPKC proteins, and a second module composed of PAR1, LGL, and SLMB proteins (*Goldstein and Macara, 2007*; *Rodriguez-Boulan and Macara, 2014*; *St Johnston and Ahringer, 2010*). Localisation of these two modules is mutually exclusive and mainly involves interplay between physical interactions and phosphorylations (*Coopman and Djiane, 2016*). aPKC phosphorylates PAR1 to exclude it from the PAR3 cortical domain. Conversely, PAR3 is phosphorylated by the kinase PAR1, recognised by the 14.3.3 proteins and excluded from the PAR1 cortical domain (*Benton and St Johnston, 2003*; *Morais-de-Sá et al., 2010*). PAR3 is also maintained at the plasma membrane through physical interactions with membrane lipids, phosphoinositides (*Krahn et al., 2010a*). We reported in particular that PI(4,5)P2 (phosphatidyl inositol 4,5 diphosphate) controls PAR3 apical targeting in epithelial cells

(*Claret et al., 2014*). In addition, the cytoskeleton seems to be important for proper localisation of PAR3, and it is noteworthy that the apical actin network is required to position PAR3 apically as well as microtubules (MTs) via dynein-dependent transport in *Drosophila* embryos (*Harris and Peifer, 2005*; *McKinley and Harris, 2012*).

In *Drosophila* oocytes, PAR3, at the anterior cortical domain, and PAR1, at the posterior, specify the polarity axes by controlling the MT organisation (*Cox et al., 2001a*; *Doerflinger et al., 2003*), and thus the localisation of determinants such as *bicoid*, *oskar*, and *gurken* mRNAs, crucial for subsequent future embryo development (*St Johnston, 2005*). Generating mutually exclusive cortical domains is especially important for polarisation of this large single-cell system. However, throughout oogenesis, the asymmetric localisation of PAR proteins is dynamic and PAR1 and PAR3 domains are not always mutually exclusive. During early oogenesis, localisation of PAR3 and PAR1 is independent (*Huynh et al., 2001*), while at mid-oogenesis (stage 7–8), PAR3 and PAR1 overlap at the posterior plasma membrane. Both show mutually exclusive localisations at stage 9 (*Doerflinger et al., 2010*).

During stages 8–10, which correspond to the critical period for localisation of the polarity axes determinants (*Steinhauer and Kalderon, 2006*), the oocyte undergoes a rapid threefold size increase while subcellular localisation of PAR modules has to remain strictly conserved. Although the mutual antagonism between PAR3/aPKC/PAR6 and PAR1/LGL/SLMB modules plays an important role in setting up PAR3 restriction (*Morais-de-Sá et al., 2014*; *Tian and Deng, 2008*), the molecular mechanisms involved in these processes remain elusive and may not be sufficient to sustain PAR3 asymmetry. Indeed, computer modelling has highlighted that mutual antagonism between the two complexes is not enough to maintain asymmetry between the polarity determinants (*Fletcher et al., 2012*). Moreover, the fate of PAR3, once phosphorylated by PAR1, and how it is redirected from the posterior to the anterior cortex is unknown. PAR3 could diffuse laterally in contact with plasma membrane until it reaches the anterior, or it could be dispersed from the posterior membrane in the cytoplasm as has been suggested for the *Drosophila* embryo (*McKinley and Harris, 2012*) and then recycled. Our previous findings have highlighted the role of the PIP5Kinase Skittles (SKTL) and its product PI(4,5)P2 in PAR3 localisation in oocytes and epithelial cells (*Claret et al., 2014*; *Gervais et al., 2008*). PI(4,5)P2, among other functions (*Tan et al., 2015*; *Zimmermann et al., 2005*), is crucial to recruit the endocytic machinery at the plasma membrane during the first step of endocytosis (*Compagnon et al., 2009*; *Posor et al., 2015*). We hypothesise that SKTL, by controlling endocytosis and/or vesicular trafficking could regulate indirectly PAR3 localisation.

Here, with use of quantitative analysis, we examined the precise evolution of PAR3 distribution during stages 8–10 of oogenesis. We show that PAR3 is excluded from the posterior cortex much later than establishment of PAR1 to the posterior and that actin cytoskeleton and endocytosis are important for this process. Subsequently, MTs and dynein motor are required for PAR3 transport to the anterior plasma membrane. This transport is connected with vesicular trafficking, cytoplasmic PAR3 being associated with PI(4,5)P2-enriched endosomes. We provide evidence of physical interactions between PAR3, SKTL, and the dynein light intermediate chain DLIC, which could explain transport of PAR3 by dynein directly on vesicles. Finally, we found that IKK-related kinase, IKKε in the correct localisation of PAR3. Knockdown of IKKε leads to an accumulation of PAR3 near the minus ends of MTs. Altogether our results point to a connection between membrane trafficking and dynein-mediated transport to sustain PAR3 asymmetry in the *Drosophila* oocyte.

## Results

### Fine tuning of PAR3 distribution along the anterior posterior axis

To characterise the evolution of PAR3 distribution during oocyte development, we developed a quantification method to monitor the precise variation of PAR3 distribution along the plasma membrane during late oogenesis (*Figure 1A*). The oocyte plasma membrane was therefore subdivided into three regions: the anterior plasma membrane (APM), which corresponds to the membrane in contact with the nurse cells, the posterior plasma membrane (PPM), which corresponds to the posterior domain of the oocyte where PAR1 and Staufen are localised (*Figure 1—figure supplement 3A–B*), and the lateral plasma membrane (LPM), which corresponds to the plasma membrane between the two previous regions (*Figure 1A*). The APM domain corresponds to juxtaposition of oocyte and nurse cell plasma membranes. Therefore, for each oocyte individually, we quantified the mean

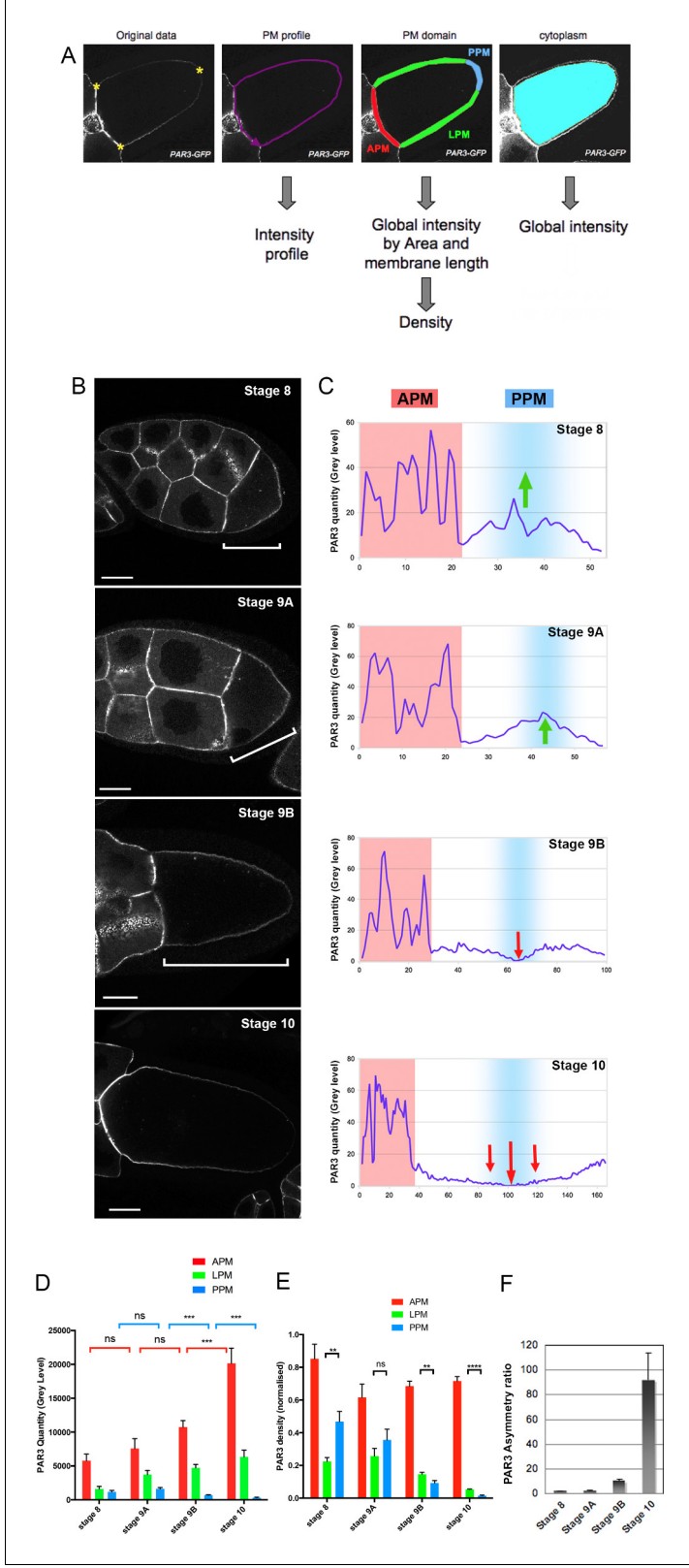

**Figure 1.** Dynamic PAR3 distribution along the oocyte anterior posterior axis. (A) Description of the quantification Fiji Macro. After selecting three points in the oocyte (yellow stars) and delimitation of oocyte perimeter, the macro allows us to obtain different data about protein repartition in oocytes: the intensity profile of the plasma membrane (magenta); the mean fluorescent intensity and the length of each of the plasma membrane domainsare

*Figure 1 continued on next page*

*Figure 1 continued*

automatically generated (anterior/APM in red; lateral/LPM in green; posterior/PPM in blue) and the mean signal intensity inside the cytoplasm (cyan). (**B–E**) Distribution of PAR3 between stages 8 and 10 in representative examples. (**B**) Localisation of PAR3-GFP, expressed in the germline under control of the maternal driver *Tub67c-GAL4*, from stage 8 to stage 10. The brackets indicate oocytes. (**C**) Representative intensity profiles of plasma membrane distribution in a single oocyte (APM in red, PPM in blue). Green arrows highlight PAR3 posterior accumulation, and red arrows PAR3 posterior exclusion. (**D**) Raw quantity of PAR3 in each plasma membrane domain from stage 8 to stage 10 (APM in red; LPM in green; PPM in blue). (**E**) To avoid size/expression fluctuations of oocytes between the different mutant genotypes, PAR3 distribution has been normalised by the length of the membrane and the total oocyte signal (density/total). In this case, we cannot compare the level between the different stages but only the asymmetrical distribution of PAR3 between the different domains. (**F**) Evolution of PAR3 asymmetry from stage 8 to stage 10. The asymmetry ratio (APM/PPM of PAR3 density) highlights the increase of PAR3 polarity in oocytes from stage 8 to stage 10. Stage 8, n = 8; stage 9A, n = 9; stage 9B, n = 15; stage 10, n = 14. Mann-Whitney test, NS: not significant; **: p<0.01 ; ***: p<0.001; ****: p<0.0001. Error bars indicate SEM. The scale bars represent 30 µm in (**B**) and in all following figures.

DOI: https://doi.org/10.7554/eLife.40212.002

The following source data and figure supplements are available for figure 1:

**Source data 1.** Quantification of TubGal4; UASp PAR3-GFP egg chambers during oogenesis.
DOI: https://doi.org/10.7554/eLife.40212.007
**Figure supplement 1.** Membrane et developmental stage definitions.
DOI: https://doi.org/10.7554/eLife.40212.003
**Figure supplement 2.** PAR3 distribution without total signal normalisation.
DOI: https://doi.org/10.7554/eLife.40212.004
**Figure supplement 3.** Control experiments of PAR3 quantification process.
DOI: https://doi.org/10.7554/eLife.40212.005
**Figure supplement 3—source data 1.** Proportion of anterior PAR3 (APM) in oocytes versus in adjacent nurse cells.
DOI: https://doi.org/10.7554/eLife.40212.006

fluorescent intensity of oocyte adjacent nurse cells. We then removed, from the anterior signal, the mean fluorescent intensity of the nurse cell plasma membrane to precisely quantify the signal coming only from the APM of this oocyte. Afterwards, the intensity profile along the plasma membranes, as well as the global intensity by plasma membrane and cytoplasm domains, were compiled and the signal density was analysed (see Materials and methods for details).

To identify and to characterise PAR3 regulation mechanisms, we first used a *Drosophila* strain expressing a Bac-encoded PAR3-GFP under its endogenous regulatory regions, in the absence of endogenous PAR3 (*Figure 1—figure supplement 1A*, (*Besson et al., 2015*)). However, accumulation of PAR3 at the apex of somatic follicular cells around the oocyte (*Figure 1—figure supplement 1B*) masks the potential localisation of PAR3 at the plasma membrane of oocyte, with the two membranes being very close. To circumvent this difficulty, we chose to follow PAR3 with a GFP tag, expressed only in the oocyte and its associated nurse cells (*Benton and St Johnston, 2003*). In this context, as previously reported (*Doerflinger et al., 2010*), PAR3 mostly accumulates at the anterior plasma membrane (*Figure 1B–C*). With our quantitative tool, we measured PAR3 signals coming from the different plasma membrane subdomains during late oogenesis (*Figure 1D*). We observe a clear increase of PAR3 quantity at the APM (in red) from stage 8 to stage 10, and in parallel an exclusion of PAR3 on the PPM (in blue). Thereafter, to avoid fluctuations associated with membrane growth during oogenesis or with experimental procedures, we normalised the raw data to the total amount of PAR3 signal in each oocyte and related to the membrane length (cf. materiel and methods, and *Figure 1—figure supplement 2*). With these density results, although we cannot compare the quantities between different stages, we can follow the evolution of asymmetry in a stage. As expected, PAR3 density is highest at the oocyte APM (*Figure 1E*). However, PAR3 density repartition presents a striking dynamic in both PPM and LPM. At stage 8 and early stage 9 (stage 9A, *Figure 1—figure supplement 1C*), PAR3 is denser at the PPM than at the LPM (*Figure 1C*, green arrows). Then at late stage 9 (stage 9B, *Figure 1—figure supplement 1C*), PAR3 is progressively excluded from the PPM (*Figure 1C*, red arrows). These changes in PAR3 distribution reflect establishment of two distinct plasma membrane domains and are highlighted by the asymmetric ratio of

PAR3 measured by the anterior to posterior density ratio (*Figure 1F*). The beginning of PAR3 posterior exclusion in the middle of stage 9 correlates with the localisation switch of other factors such as Staufen (*Figure 1—figure supplement 3B*) toward the posterior pole of the oocyte and correlated to the MT network reorganisation (*Januschke et al., 2006*). This result is surprising as PAR1 is assumed to exclude PAR3 from the plasma membrane, yet PAR1 is already localised at the posterior pole from at least stage 7, long before PAR3 exclusion (*Doerflinger et al., 2010*). This may indicate that other processes participate in the disappearance of PAR3 from the PPM and the LPM. As the LPM appeared to follow the PPM comportment, thereafter we focused on APM accumulation and on PPM exclusion, two mechanisms important for establishment of antero-posterior polarity.

## Posterior exclusion and anterior accumulation are two PAR3 localisation processes that can be decoupled/separated

To understand how PAR3 is excluded from the posterior domain and enriched at the anterior domain in the oocyte, we investigated further cytoskeleton involvement in this process. With latrunculin drug treatment, PAR3 is still predominantly at the APM as under control conditions (*Figure 2A and C*, *Figure 2—figure supplement 1*). However, its posterior exclusion, which normally occurs at stage 9B, is not observed (*Figure 2B*), indicating that the actin network is required for PAR3 posterior exclusion. We next addressed the potential MT requirement for PAR3 antero/posterior distribution in the oocyte. In the presence of colchicin, a drug that depolymerises MT, PAR3-polarised distribution along the antero/posterior axis is lost and tends toward isotropy (*Figure 2A and C*, *Figure 2—figure supplement 1*). Compared to control, PAR3 APM localisation is strongly reduced (*Figure 2C*). However, PAR3 is still excluded from the PPM although to a lesser extent than the control (*Figure 2A–B*). This confirms the importance of MTs in PAR3-polarised localisation, except at the posterior region where exclusion is still present. Hence it appears that in the oocyte, posterior exclusion and anterior accumulation can be uncoupled. We also note that both MT and actin disassembly increase cytoplasmic accumulation of PAR3 in dotted structures (*Figure 2D*). We verified that these dotted structures are also detected in the absence of PAR3 overexpression with a PAR3-Protein trap strain (*Januschke and Gonzalez, 2010*). In this condition, we can observe some dotted structures in the oocyte cytoplasm (*Figure 2—figure supplement 2A*). Furthermore, when microtubules are depolymerised by colchicin, PAR3 accumulates in numerous dotted structures in the cytoplasm (*Figure 2—figure supplement 2B–D*).

## By producing PI(4,5)P2, SKTL controls anterior accumulation and posterior exclusion of PAR3

PAR3 is a cytoplasmic protein that can interact with membranes by direct interaction with phospholipids, in particular PI(4,5)P2 (*Claret et al., 2014*; *McKinley et al., 2012*; *Wu et al., 2007*), and/or by interaction with membrane-associated proteins such as those of the adherens junction (*Coopman and Djiane, 2016*).

We have previously shown that the PIP5Kinase SKTL, by providing the phosphoinositide PI(4,5)P2, is crucial to maintaining PAR3 at the adherens junctions in epithelial cells (*Claret et al., 2014*). In oocytes, PAR3 is associated with the plasma membrane (*Gervais et al., 2008*) but also forms some particles in the cytoplasm. These particles are also associated in part with PI(4,5)P2-containing membranes (*Figure 3G*) and the lipid kinase SKTL, which produces PI(4,5)P2 (*Figure 3H*). By immunostaining and by colocalisation using Mander's overlap coefficient, we identified that half of PAR3 vesicles ($50.05\% \pm 0.08$ SEM, n = 8) are PIP(4,5)P2 -positive (*Figure 3G*).

We then investigated SKTL requirements on PAR3 polarised distribution along the oocyte anterior posterior axis focusing both on the amount of SKTL and on its kinase activity. In the absence of SKTL ($sktl^{2.3}/sktl^{\Delta 5}$), PAR3 is accumulates more at the PPM than at the APM (*Figure 3A, D and E*). Conversely, when SKTL is overexpressed (OE), PAR3 is more excluded from the PPM than under control conditions (*Figure 3B, D and E*). Thus, SKTL seems to have a preponderant function to exclude PAR3 from the PPM and to increase PAR3 density at the APM. To monitor whether SKTL kinase activity is required for this process, we performed the same experiment with a kinase dead form, SKTL[DNRQ] whose mutation in mammalian homolog leads to a dominant negative effect (*Coppolino et al., 2002*). In such a case, SKTL[DNRQ] overexpression does not enhance PAR3 exclusion from the PPM (*Figure 3C, D and E*). PAR3 distribution along the anterior posterior axis tends to be

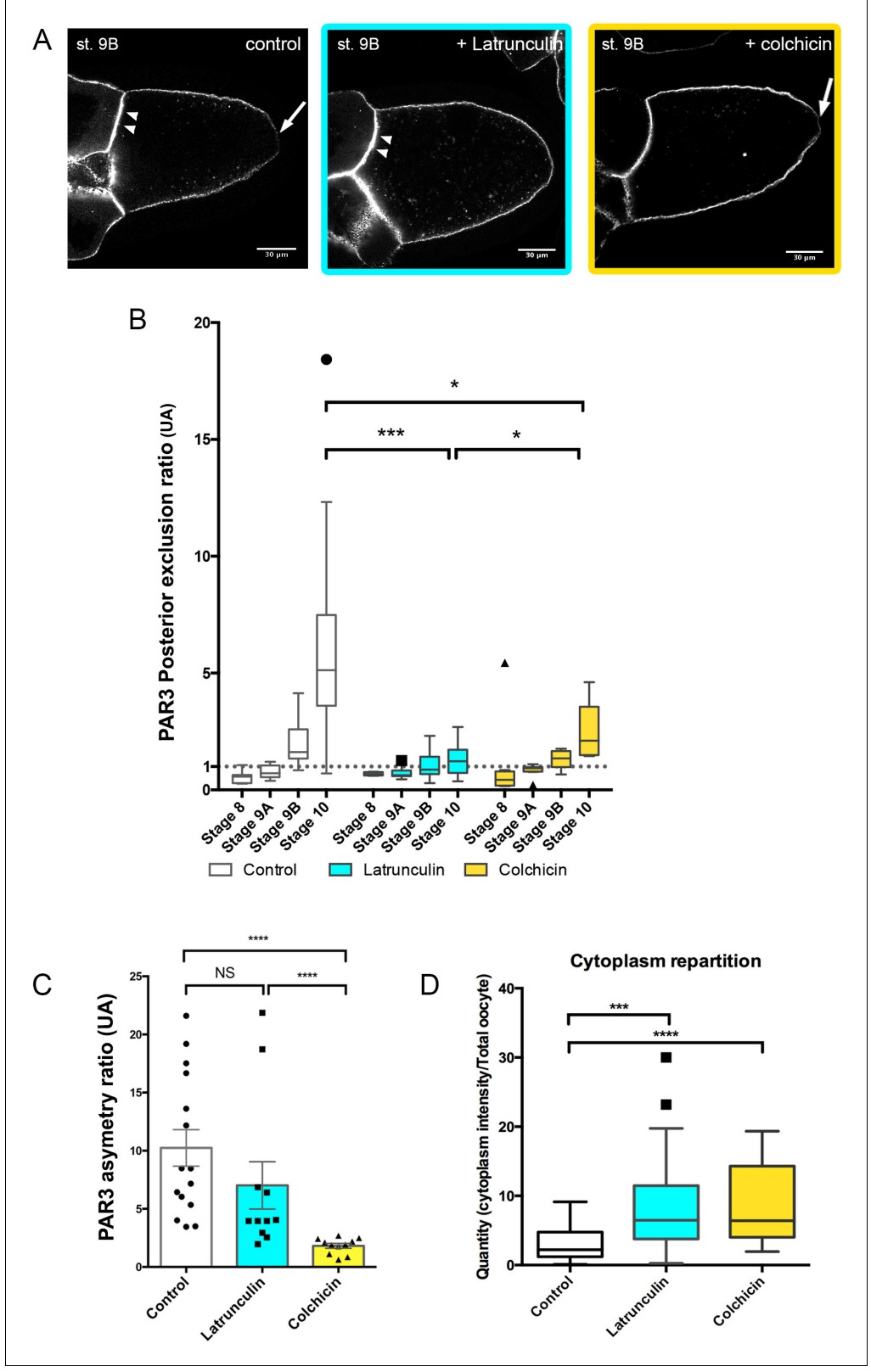

**Figure 2.** Cytoskeleton involvement in PAR3 polarity. (**A–D**) Role of cytoskeleton on PAR3 distribution. Flies are nourished with latrunculin (cyan) for 48 h, colchicin (yellow) for 24 h, or only yeast paste (control). (**A**) Representative distribution of PAR3-GFP in stage 9B oocytes. Note the PPM exclusion of PAR3 (arrow) or the strong APM accumulation (arrowhead). (**B**) The PAR3 posterior exclusion ratio (ratio LPM/PPM density) is
*Figure 2 continued on next page*

*Figure 2 continued*
represented between stage 8 and stage 10 oocytes. Under the value of 1, there is a posterior accumulation of PAR3, and above 1 a posterior exclusion. (C) Antero-posterior asymmetry (ratio APM/PPM density at stage 9B) is strongly affected by colchicin, but not by latrunculin. (D) The two drugs lead to an increase in the cytoplasmic fraction of PAR3 in stage 9B oocytes. Mann-Whitney test, NS: not significant; *$p < 0.05$ ; ***$p < 0.001$ ; ****$p < 0.0001$. Error bars indicate SEM. Control (stage 8, n = 8; stage 9A, n = 9; stage 9B, n = 15; stage 10, n = 14);+latrunculin (stage 8, n = 5; stage 9A, n = 12; stage 9B, n = 11; stage 10, n = 9);+colchicin (stage 8, n = 8; stage 9A, n = 9; stage 9B, n = 11; stage 10, n = 6).
DOI: https://doi.org/10.7554/eLife.40212.008

The following source data and figure supplements are available for figure 2:

**Source data 1.** PAR3 posterior exclusion ratio in oocyte of flies fed with latrunculin, colchicin, or without drugs.
DOI: https://doi.org/10.7554/eLife.40212.013
**Source data 2.** PAR3 asymmetry ratio in oocyte of flies fed with latrunculin, colchicin, or without drugs.
DOI: https://doi.org/10.7554/eLife.40212.014
**Source data 3.** PAR3 quantity (cytoplasm intensity/total oocyte intensity) in oocyte of flies fed with latrunculin, colchicin, or without drugs.
DOI: https://doi.org/10.7554/eLife.40212.015
**Figure supplement 1.** Cytoskeleton involvement in PAR3 distribution.
DOI: https://doi.org/10.7554/eLife.40212.009
**Figure supplement 1—source data 1.** Quantification of PAR3-GFP density (normalised) between stage 8 and stage 10 upon latrunculin and colchicin treatment.
DOI: https://doi.org/10.7554/eLife.40212.010
**Figure supplement 2.** Effect of colchicin on distribution of PAR3 present at an endogenous level.
DOI: https://doi.org/10.7554/eLife.40212.011
**Figure supplement 2—source data 1.** Density of cytoplasmic fraction of PAR3-GFP (trap line) at stage 9B upon latrunculin and colchicin treatment.
DOI: https://doi.org/10.7554/eLife.40212.012

isotropic (*Figure 3C and D*). We can conclude that SKTL kinase activity, hence the production of PI (4,5)P2, is essential to regulate PAR3 distribution.

## SKTL can bypass PAR1-dependent posterior exclusion

Upon phosphorylation, PAR1 excludes PAR3 from the posterior domain of the oocyte (*Benton and St Johnston, 2003*). Accordingly, with our quantification method, we found that PAR3 was not excluded from the PPM upon PAR1 knockdown by RNAi (*Figure 4B and D* and *Figure 4—figure supplement 1*). Likewise, a PAR3-AA mutant form, non phosphorylable by PAR1 (*Benton and St Johnston, 2003*), is not excluded from the PPM (*Figure 4C–D*). Thus, PAR1 through its role in PAR3 phosphorylation is required to exclude PAR3 from the PPM. As SKTL also controls the posterior exclusion of PAR3, we combined the knockdown of PAR1, which normally presents no PAR3 exclusion, with the overexpression of SKTL, which increases the exclusion. In this case, we observed that SKTL can still exclude PAR3 even in the absence of PAR1 (*Figure 4E*). We then confirmed this observation using the PAR3-AA mutant form (*Figure 4E*). In this case too, SKTL overexpression can still exclude PAR3 from the posterior membrane. These results suggest that SKTL leads to PAR3 exclusion independently of PAR3 phosphorylation by PAR1.

## RAB5-dependent endocytosis is important for PAR3 PPM exclusion

To understand how SKTL could induce the PPM exclusion of PAR3, we investigate the implication of endocytosis in this process. PI(4,5)P2 has numerous functions in the cell, one of which is a role in the first steps of endocytosis (for review *Posor et al., 2015*). In *Drosophila* oocyte, PIP(4,5)P2 is required for endocytic-vesicle formation and the small GTPase RAB5 is required for maturation of these early endocytic vesicles (*Compagnon et al., 2009*).

As we previously described, in addition to a plasma membrane localisation, PAR3 is also detected in dotted structures. By immunostaining and by colocalisation using Mander's overlap coefficient, we identified that half of PAR3 vesicles (50.2% ± 0.05 SEM, n = 10) are RAB5-positive, an early endosome marker (*Figure 5A*). Moreover, when RAB5 activity is impaired by expression of a dominant

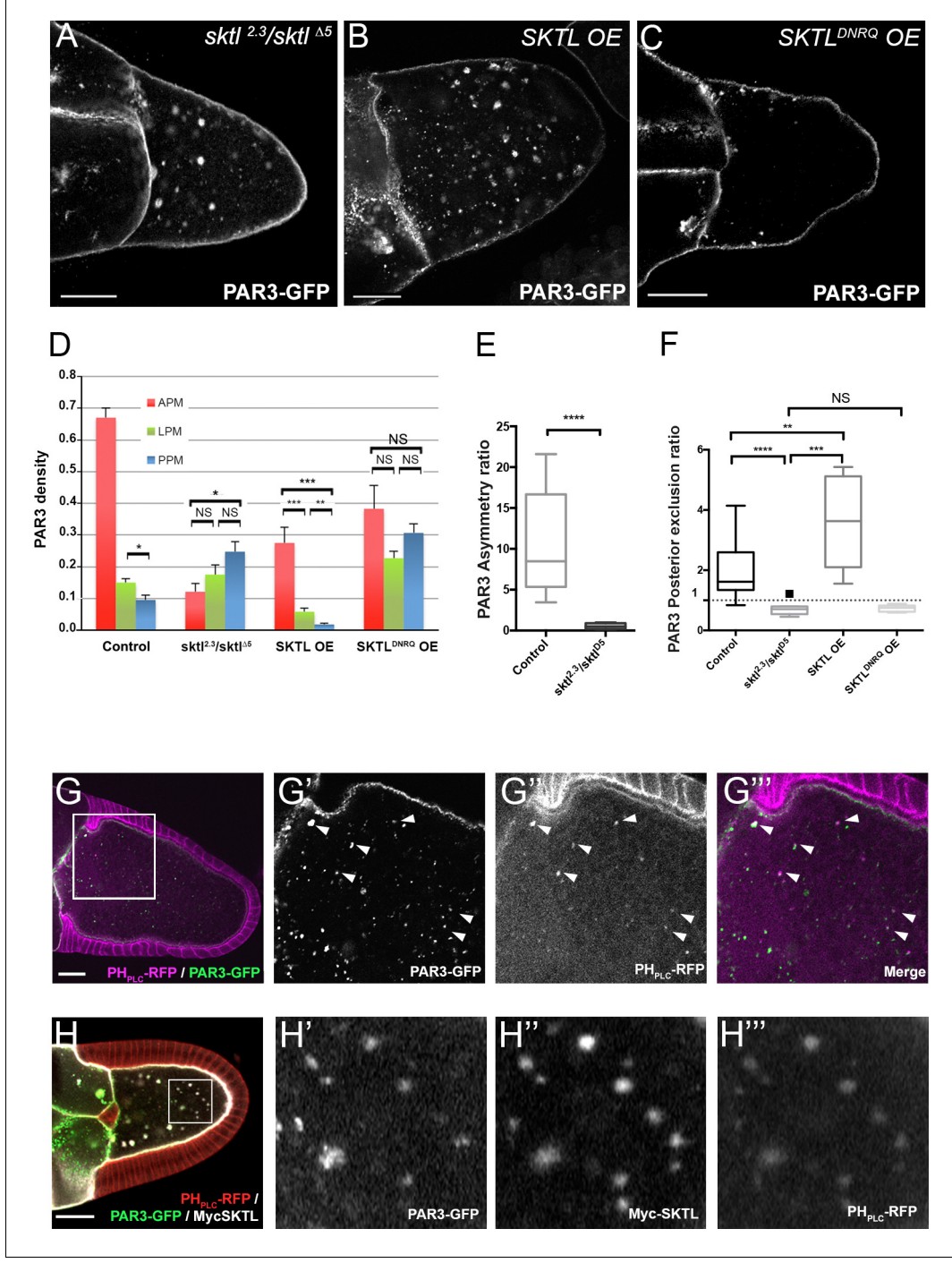

**Figure 3.** By producing PI(4,5)P2, SKTL controls PAR3 APM accumulation and PPM exclusion. (**A–F**) Distribution of PAR3 in response to SKTL activity. PAR3-GFP is expressed in germinal cells at stage 9B under control conditions, or with overexpression (OE) of Myc-SKTL, Myc-SKTL$^{DNRQ}$, or in context $sktl^{2.3}/sktl^{\Delta 5}$. (**A–C**) Representative distribution of PAR3-GFP in oocytes in these different genetic contexts. (**D**) Quantification of PAR3 density at each plasma membrane domain in $sktl$ mutant or SKTL overexpressed (OE) contexts in stage 9B oocytes. Error bars indicate SEM. (**E**) Antero-posterior asymmetry of PAR3 (ratio APM/PPM density) at stage 9B in control or $sktl$ mutant. (**F**) Quantification of PAR3 posterior exclusion ratio in $sktl$ mutant or SKTL overexpressed contexts at stage 9B. For (**D–F**): control (stage 9B, n = 10); $sktl^{2.3}/sktl^{\Delta 5}$ (stage 9B, n = 8); Myc-SKTL (stage 9B, n = 8); Myc-SKTL$^{DNRQ}$ (stage 9B, n = 10). Mann-Whitney test, NS: not significant; *p<0.05 ; **p<0.01 ; ***p<0.001 ; ****p<0.0001. (**G**) PAR3-GFP (green, **G'**) expressed in the germline is present occasionally on vesicles containing PI(4,5)P2 visualised with PH$_{PLC}$ RFP (magenta). Arrowheads show the vesicles that are associated with PAR3 and PI(4,5)P2. G', G'' and

*Figure 3 continued on next page*

*Figure 3 continued*

G''' are magnifications of G (white frame). (**H**) Colocalisation of PAR3 (green, **H'**), PI(4,5)P2 visualised with PH$_{PLC}$ RFP (red, **H'''**), and SKTL visualised with myc tag (white, **H''**). Note that coexpression of PAR3 and SKTL increase the cytoplasmic dotted localisation of PAR3. Scale bars indicate 30 μm.

DOI: https://doi.org/10.7554/eLife.40212.016

The following source data is available for figure 3:

**Source data 1.** Quantification of PAR3 density (normalised) at each plasma membrane domain in sktl mutant or SKTL overexpressed (OE) contexts in stage 9B oocytes.

DOI: https://doi.org/10.7554/eLife.40212.017

**Source data 2.** Quantification of PAR3 asymmetry ratio in sktl mutant or control at stage 9B.

DOI: https://doi.org/10.7554/eLife.40212.018

**Source data 3.** Quantification of PAR3 posterior exclusion ratio in sktl mutant or SKTL overexpressed contexts at stage 9B.

DOI: https://doi.org/10.7554/eLife.40212.019

negative form, RAB5$^{S43N}$ (*Entchev and González-Gaitán, 2002*), as revealed by lipophilic dye uptake, the endocytosis is strongly reduced at the posterior of the oocyte (*Figure 5—figure supplement 1A–B*) and PAR3 distribution is affected (*Figure 5D and E*). The same result is observed using a RAB5 knockdown (*Figure 5C and E*, RAB5 RNAi). PAR3 accumulation at the APM is lost and its density significantly increases at the PPM, leading to loss of its PPM exclusion (*Figure 5E and F*). Thus, proper PAR3 distribution along the antero-posterior axis appears to rely on endocytosis being removed from the PPM and being enriched at the APM.

## RAB11-dependent recycling pathway is required for PAR3 APM enrichment

To determine the nature of all the PAR3 dotted structures that can be detected in the wild type context, we immunostained different compartments of the cell and measured their colocalisation with PAR3 (*Figure 6A*).

PAR3-associated vesicles are enriched in PI(4,5)P2, and colocalise with the RAB5-positive compartment (early endosome), but also the RAB11-positive compartment (26.8% (±0.06 SEM, n = 13) of cytoplasmic PAR3 colocalises with RAB11-recycling endosomes *Figure 6A and B*). There is no colocalisation with endoplasmic reticulum (KDEL), Golgi compartment (SYX16), or late endosome (HRS) markers (*Figure 6A* and *Figure 6—figure supplement 1*). Taken together, these results indicate that PAR3 can be associated with RAB5 and RAB11 endosomes. However, endosomes could be a mosaic of several RAB domains present in continuity on the same membrane vesicle (*Sönnichsen et al., 2000*; *Wandinger-Ness and Zerial, 2014*). Here we do not know if PAR3 is associated with a unique endosome supporting RAB5 and RAB11 or with a specific compartment containing only one RAB.

To identify whether RAB11 is involved in asymmetric distribution of PAR3, we knocked down RAB11 in the oocyte through germline clones with *rab11*$^{p2148}$ allele, which precludes oocyte development (*Jankovics et al., 2001*) and monitors PAR3 distribution (*Figure 6F*). There was no significant difference on PAR3 PPM exclusion between control (*Figure 6D–E*) and *rab11*$^{p2148}$ mutant clones (*Figure 6D and F*). However, we noted a RAB11 effect on PAR3 APM enrichment (*Figure 6C*). Thus RAB11 seems to be more important for APM accumulation than for the PPM exclusion.

Finally, we monitored whether PAR3, which comes from the nurse cells through the ring canals, is also connected to membrane traffic. We noted that PAR3 accumulates in the vicinity of the ring canals and that it is associated with PI(4,5)P2 membrane and with RAB11 endosomes (*Figure 6—figure supplement 2*). This suggests that neo-synthesised PAR3 is also transported in the oocyte in association with endosomes.

## DLIC, PAR3, and SKTL are interacting partners

To uncover new interacting partners for SKTL upon polarity establishment of PAR3, we performed a proteomic screen by immunoprecipitation and then mass spectrometry analysis. We identified, after

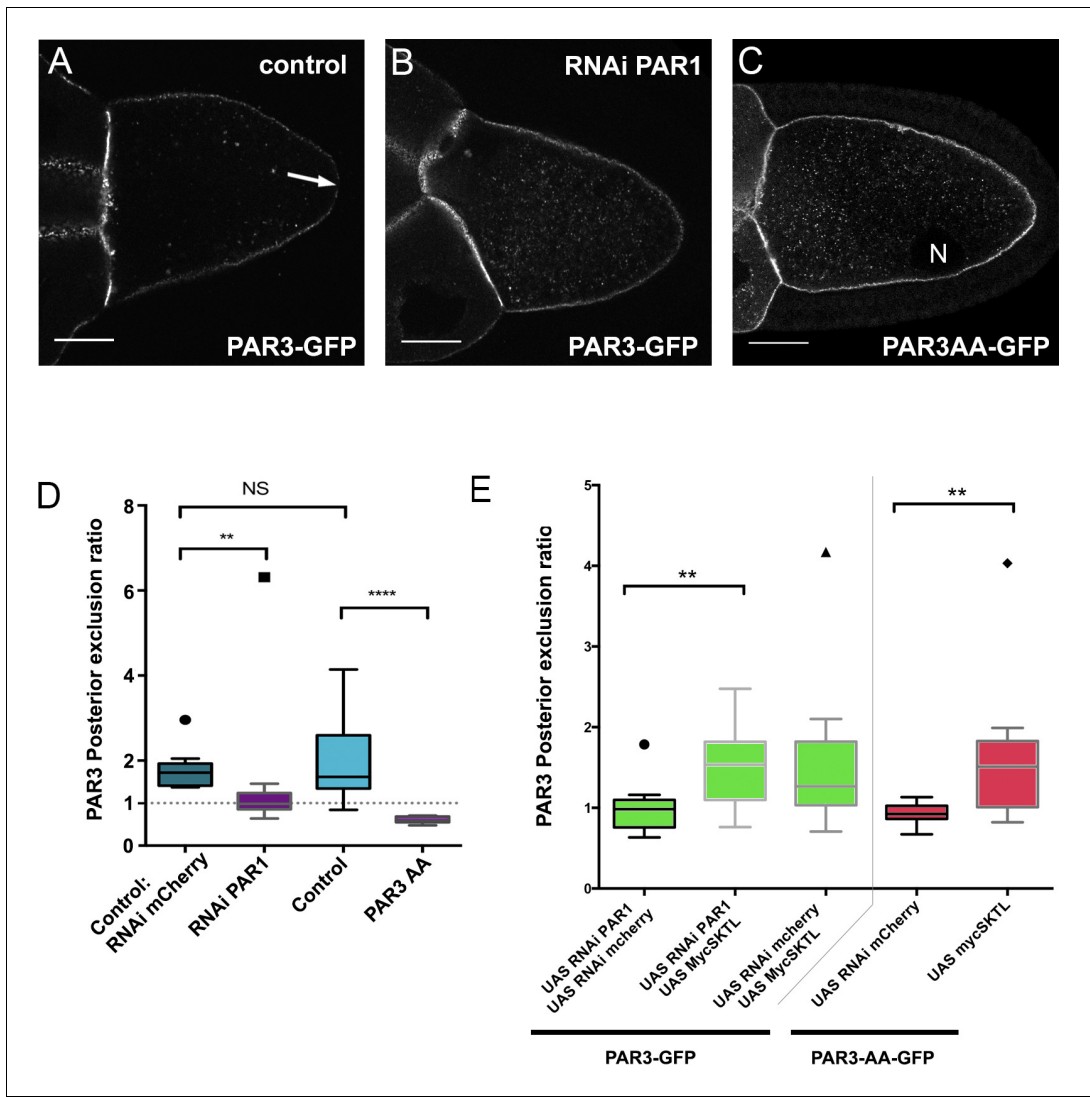

**Figure 4.** SKTL-dependent PAR3 posterior exclusion bypasses regulation by PAR1. (**A–C**) Representative distribution of PAR3-GFP in stage 9B oocyte in a control situation (**A**), in RNAi PAR1 context (**B**), or when PAR3 phosphorylation sites by PAR1 are mutated (**C**). In the control genotype, PAR3 is excluded from PPM (arrow), unlike the other two genotypes. N indicates the oocyte nucleus position. The scale bars represent 30 μm. (**D**) PAR3 posterior exclusion in response to PAR1 at stage 9B. In germinal cells, PAR3AA-GFP, a mutant form, non phosphorylable by PAR1 or PAR3-GFP is expressed with nothing, with *par1,* or with *mCherry* knock-down contexts. The posterior exclusion ratios in stage 9B oocytes are represented. PAR3 (stage 9B, n = 15); PAR3-AA (stage 9B, n = 6); PAR3 RNAi mCherry (stage 9B, n = 10); PAR3 RNAi PAR1 (stage 9B, n = 10). (**E**) SKTL effect on PAR3 posterior exclusion is observed in combination with a PAR1 activity decrease (in green) or with the PAR3-AA non phosphorylable form (in red). PAR3, RNAi PAR1, RNAi mCherry (stage 9B, n = 10); PAR3, RNAi PAR1, mycSKTL (stage 9B, n = 11) PAR3 RNAi PAR1 (stage 9B, n = 10); PAR3, RNAi mCherry, mycSKTL (stage 9B, n = 10); PAR3-AA, RNAi mCherry (stage 9B, n = 10); PAR3-AA, mycSKTL (stage 9B, n = 10). Mann-Whitney test, NS: not significant; *p<0.05 ; **p<0.01 ; ***p<0.001 ; ****p<0.0001.
DOI: https://doi.org/10.7554/eLife.40212.020

The following source data and figure supplement are available for figure 4:

**Source data 1.** Quantification of PAR3 posterior exclusion in response to PAR1 at stage 9B.
DOI: https://doi.org/10.7554/eLife.40212.022

**Source data 2.** Quantification of PAR3 posterior exclusion in response to PAR1 at stage 9B in combination with SKTL.
DOI: https://doi.org/10.7554/eLife.40212.023

**Figure supplement 1.** Validation of RNAi PAR1 efficiency.
DOI: https://doi.org/10.7554/eLife.40212.021

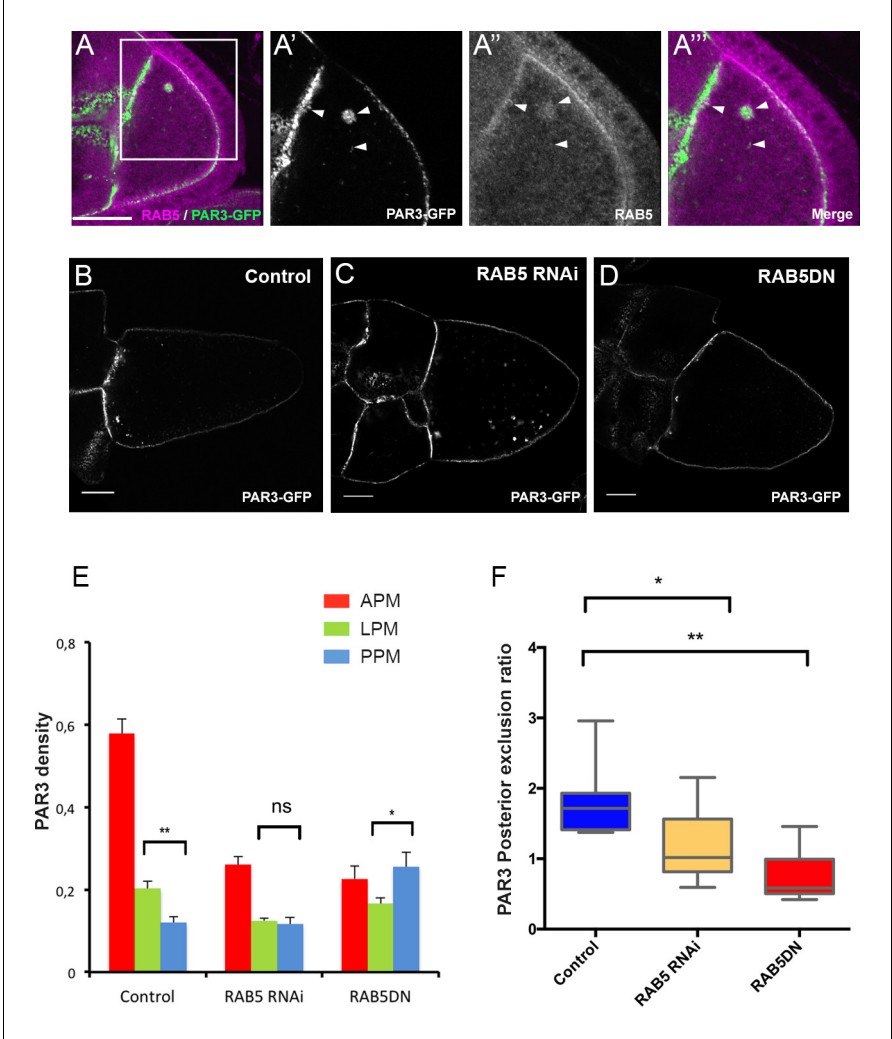

**Figure 5.** PAR3 asymmetry depends on RAB5. (**A**) PAR3-GFP (**A'**), (**A'''** green) expressed in germline is present occasionally in RAB5-positive early endosomes (**A''**), (**A'''**) magenta). A', A'' and A''' are magnifications of A (white frame). Arrowheads show the vesicles that are associated with PAR3 and RAB5. (**B–F**) PAR3 distribution in response to RAB5 activity impairment. PAR3-GFP is expressed in germinal cells at stage 9B in RAB5 RNAi, RAB5DN[(S43N)] or in *mCherry* knockdown (control) contexts. (**B–D**) Representative images of PAR3 distribution under control conditions (**B**), in RAB5 RNAi (**C**), or in RAB5DN[(S43N)] (**D**). Scale bars indicate 30 μm. (**E**) Quantification of PAR3 density at each plasma membrane domain. Error bars indicate SEM. (**F**) Quantification of PAR3 posterior exclusion in different RAB5 mutant contexts. Control (stage 9B, n = 10); RAB5 RNAi (stage 9B, n = 12); RAB5DN[(S43N)] (stage 9B, n = 6). Mann-Whitney test, ns: not significant; *p<0.05 ; **p<0.01.
DOI: https://doi.org/10.7554/eLife.40212.024

The following source data and figure supplement are available for figure 5:

**Source data 1.** Quantification of PAR3 density at each plasma membrane domain of stage 9B oocytes in response to RAB5 activity impairment.
DOI: https://doi.org/10.7554/eLife.40212.026

**Source data 2.** Quantification of PAR3 posterior exclusion in different RAB5 mutant contexts at stage 9B.
DOI: https://doi.org/10.7554/eLife.40212.027

**Figure supplement 1.** Validation of RAB5DN effect on endocytosis.
DOI: https://doi.org/10.7554/eLife.40212.025

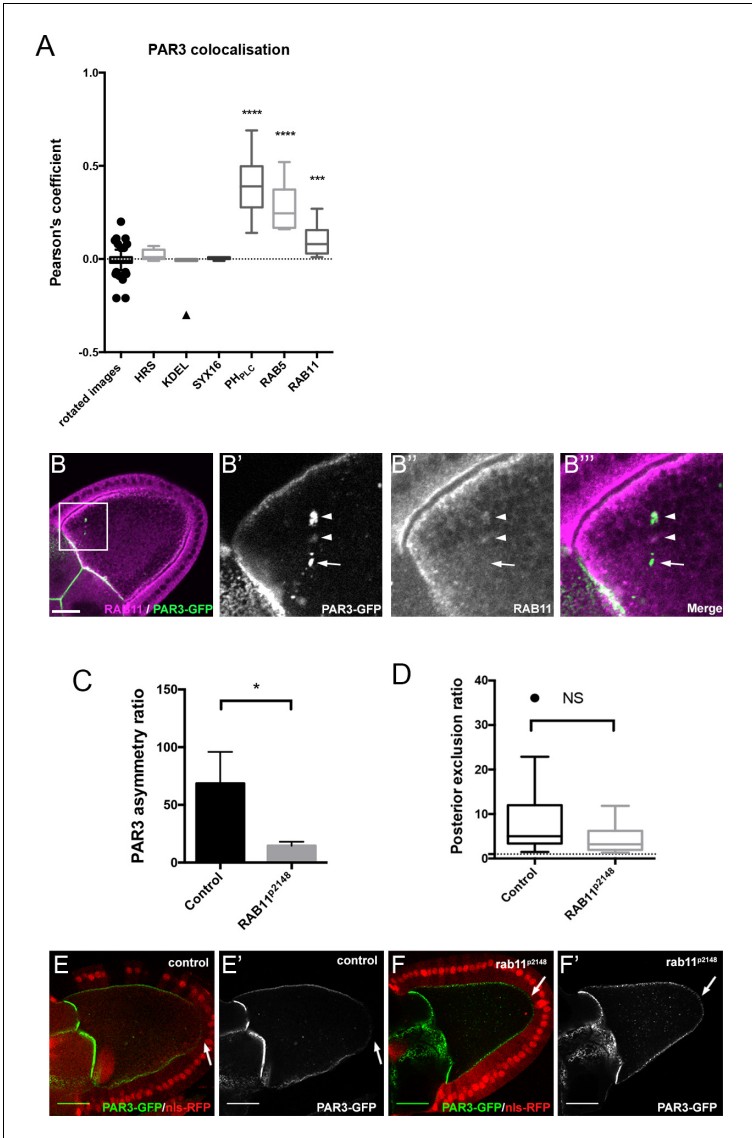

**Figure 6.** Role of RAB11 on PAR3 asymmetrical localisation. (**A**) Colocalisation of PAR3 (PAR3-GFP maternally expressed) with vesicular trafficking markers. To quantify the colocalisation, we measured the Pearson's coefficient on stage 9B oocytes and results are presented in the box plot. As a control, we used the value of colocalisation obtained with the same images but after rotation of one by 90 degrees. The stars represent the p-value with the control. For HRS, KDEL, and SYX16: n = 7; for $PH_{PLC}$: n = 8; for RAB5: n = 10; for RAB11: n = 13. (**B**) PAR3-GFP (**B'**), green) expressed in germline is present occasionally in RAB11-positive recycling endosomes (**B''**), (**B'''**) magenta). **B'**, **B''**, and **B'''** are magnifications of **B** (white frame). Arrowheads show the vesicles associated with PAR3 and RAB11. The arrow points to the vesicle associated only with PAR3. (**C,D**) RAB11 effect on asymmetrical distribution of PAR3. The antero-posterior asymmetry (**C**) and the posterior exclusion (**D**) of PAR3 at stage 9B have been evaluated in $rab11^{P2148}$ germline clones. Control (stage 9B, n = 10); $rab11^{P2148}$ (stage 9B, n = 10). Mann-Whitney test, NS: not significant; *p<0.05; ***p<0.001 ; ****p<0.0001. Error bars indicate SEM. (**E,F**) Representative images of PAR3 distribution in $rab11^{P2148}$ clones. The $rab11^{P2148}$ mutant cells are indicated by the absence of nuclear RFP staining (nls-RFP). (**E**) The heterozygote egg chamber ($rab11^{P2148}$/+) is used as a control. (**F**) $rab11^{P2148}$ mutant oocyte. Note that PAR3 is always excluded from PPM (arrow). Scale bars indicate 30 µm.

DOI: https://doi.org/10.7554/eLife.40212.028

The following source data and figure supplements are available for figure 6:

**Source data 1.** Quantification of PAR3 colocalisation with vesicular compartment.
DOI: https://doi.org/10.7554/eLife.40212.031

**Source data 2.** Quantification of PAR3 asymmetry ratio in rab11 mutant clones.

*Figure 6 continued on next page*

*Figure 6 continued*

DOI: https://doi.org/10.7554/eLife.40212.032

**Source data 3.** Quantification of PAR3 posterior exclusion ratio in rab11 mutant clones.

DOI: https://doi.org/10.7554/eLife.40212.033

**Figure supplement 1.** PAR3 and vesicular compartments.

DOI: https://doi.org/10.7554/eLife.40212.029

**Figure supplement 2.** PAR3 colocalises with PI(4,5)P2 and RAB11 in front of the ring canal.

DOI: https://doi.org/10.7554/eLife.40212.030

immunoprecipitation of SKTL-GFP, a peptide of the dynein light intermediate chain (DLIC), which is absent from the control. DLIC is an essential subunit of the MT-based dynein motor (*Reck-Peterson et al., 2018*). This interaction between SKTL and DLIC was confirmed by western blot with anti-DLIC antibody (*Figure 7E*).

Interestingly, LIC2, DLIC mammalian homologue, interacts with PAR3 through N-terminal dimerisation and PDZ1 domains of PAR3 (*Schmoranzer et al., 2009*). Thus, we investigated whether this interaction is conserved in *Drosophila*. We immunoprecipitated DLIC-GFP in ovarian extract and identified a weak association with endogenous PAR3 (*Figure 7D*). Interestingly, it seems that we coimmunoprecipitate predominantly a high molecular weight PAR3 form (*Figure 7—figure supplement 2*). This weight could correspond to an oligomeric form of PAR3 (*Kullmann and Krahn, 2018*).

As DLIC interacts with PAR3 and SKTL, we wondered whether an interaction between PAR3 and SKTL also occurs. Through immunoprecipitation, we found a physical interaction between PAR3 and SKTL (*Figure 7E*). Hence, as described for other PI(4,5)P2 effectors (*Choi et al., 2015*), PAR3 seems to form a tripartite complex with PI(4,5)P2 and SKTL, the enzyme that produces it (*Figure 3H*).

## Dynein regulates transport of PAR3 to the anterior plasma membrane but has only a modest impact on posterior exclusion

We have shown that MTs are important to asymmetric localisation of PAR3 and that PAR3 interacts with DLIC. In oocytes (stage 8–10), the MT network forms a gradient following the antero/posterior axis (*Januschke et al., 2006*). As MTs are more abundant and more nucleated (minus ends) at the anterior pole of oocytes (*Khuc Trong et al., 2015*; *Parton et al., 2011*), dynein, as a minus end directed motor, could be necessary for PAR3 accumulation at the APM. To investigate this possibility, we therefore knocked down dynein by expression of ShRNAs directed against the dynein heavy chain isoform (DHC64c) in the oocyte (*Sanghavi et al., 2013*). Under these conditions, PAR3 accumulation is lost at the APM and its level significantly increases at the LPM (*Figure 7A–B*). However, PAR3 is still excluded from PPM (*Figure 7C*), consistent with the weak MT requirement for this process (*Figure 2B*). Taken together, these data indicate that PAR3 could be transported in complex with the dynein motor on MTs toward the APM.

We noted that, upon Dhc64C inactivation, PAR3 is not evenly distributed along the plasma membrane but is retained in the dotted structures as shown in *Figure 7A* and *Figure 7—figure supplement 1A*. PAR3 is still associated with PI(4,5)P2 vesicles when MTs are impaired in the presence of colchicin (*Figure 7—figure supplement 1B*). Thus, PAR3 transport to the APM depends on MTs, but the association of PAR3 with vesicles does not.

## Connection between posterior exclusion and anterior accumulation of PAR3

Our results indicate that PAR3 oocyte distribution relies on two separate processes: an exclusion from the posterior plasma membrane and an accumulation at the anterior plasma membrane. Thus, we investigated whether the two processes were connected. We monitored PAR3 dynamics in the oocyte through fluorescence recovery after photobleaching (FRAP) of PAR3-GFP. As a substantial proportion of the PAR3 that accumulates at the anterior part of the oocyte could come from synthesis and subsequent transport from the nurse cells through the ring canals (*Figure 6—figure supplement 2* and *Doerflinger et al., 2010*), we wanted to dissociate this PAR3 arrival from the potential relocalisation of PAR3 in the oocyte from the posterior. To do so, we chose to photobleach PAR3-GFP localised in the nurse cells and at the APM (*Figure 8A*), at stage 9A (*Video 1*). We then followed evolution of PAR3-GFP fluorescence both at the APM and at the PPM (*Figure 8A*). In this

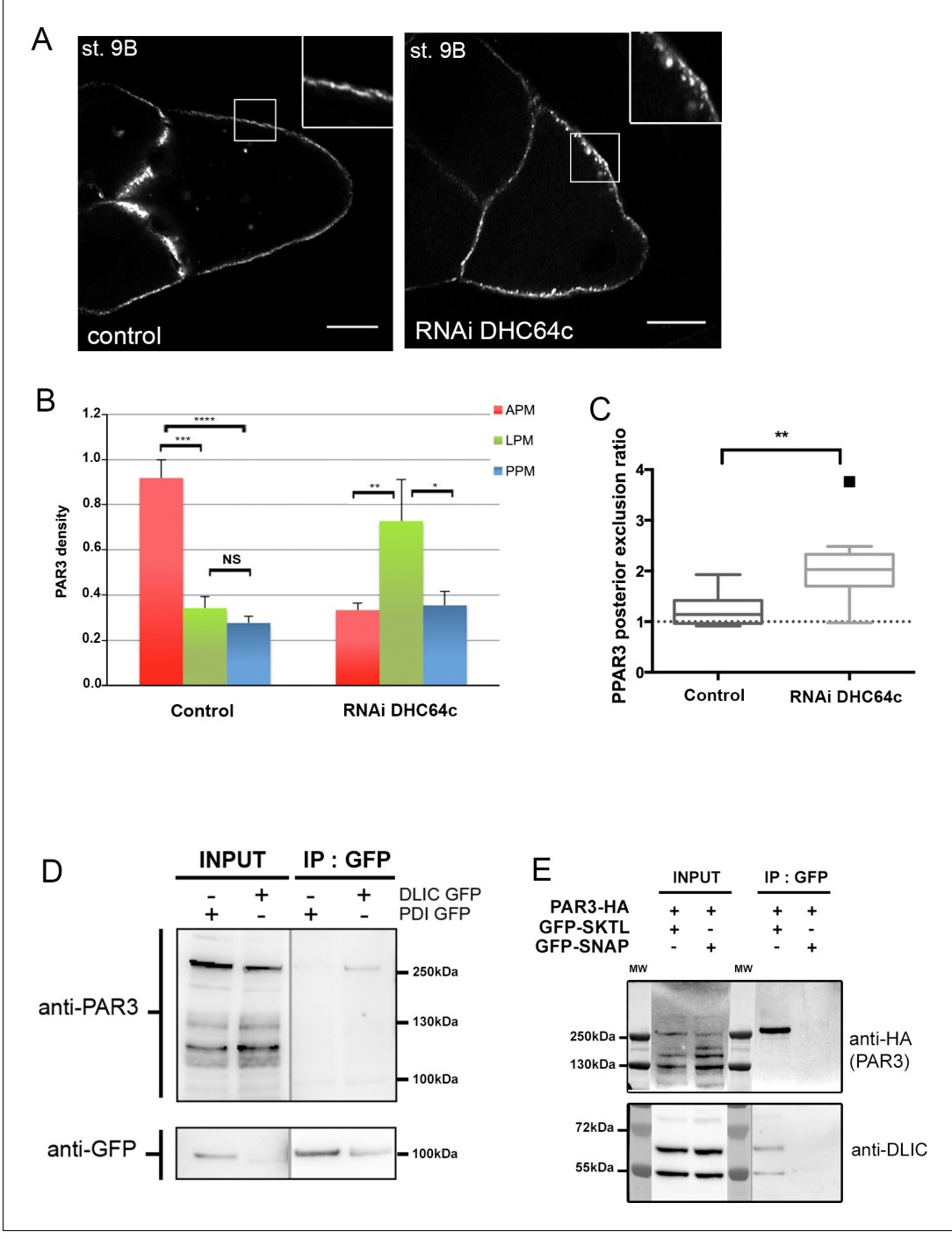

**Figure 7.** Dynein regulates PAR3 asymmetry. (**A–C**) Distribution of PAR3 in response to dynein activity decrease. PAR3-GFP is expressed in germinal cells at stage 9B in control (*osk-GAL4*; *UASp PAR3-GFP*) or *dhc64* knockdown contexts (*osk-GAL4*; *UASp RNAi dhc64*; *UASp PAR3-GFP*). (**A**) Representative distribution of PAR3 in oocyte (note the dotted accumulation of PAR3 under the plasma membrane in the insert). The scale bars represent 30 μm. (**B**) Quantification of PAR3 density at each plasma membrane domain. (**C**) Quantification of PAR3 posterior exclusion (ratio LPM/PPM) in dynein mutant context. For (**A–C**): control (stage 9B, n = 10); RNAi dhc64c (stage 9B, n = 10). For (**B, C**), Mann-Whitney tests; NS: not significant; *p<0.05 ; **p<0.01; ***p<0.001; ****p<0.0001. Error bars indicate SEM. (**D**) Co-immunoprecipitation (IP) of PAR3 by DLIC-GFP in ovarian extracts. Ovaries were dissected from *Pubi-DLIC-GFP* flies or control *Pubi-PDI-GFP* flies. The ovarian extract was incubated on magnetic beads coupled with antibody anti-GFP. PAR3 was revealed by anti-PAR3 antibody. (**E**) SKTL interacts with both PAR3 and DLIC. CoIP of PAR3-HA or DLIC with GFP-SKTL in S2 cell extracts. Cells were transfected with

*Figure 7 continued on next page*

*Figure 7 continued*

GFP-SKTL or GFP-SNAP (as control) and with PAR3-HA. The cell extracts were incubated on magnetic beads with anti-GFP antibody. PAR3 was revealed by anti-HA antibody and DLIC by anti-DLIC antibody.
DOI: https://doi.org/10.7554/eLife.40212.034

The following source data and figure supplements are available for figure 7:

**Source data 1.** Quantification of PAR3 density at each plasma membrane domain in Dhc64 knockdown at stage 9B oocytes (Panel B).
DOI: https://doi.org/10.7554/eLife.40212.037

**Source data 2.** Quantification of PAR3 posterior exclusion ratio in Dhc64 knockdown at stage 9B oocytes (Panel C).
DOI: https://doi.org/10.7554/eLife.40212.038

**Figure supplement 1.** Control experiments of PAR3 antibody specificity and charge control of immunopurification of GFP-SKTL.
DOI: https://doi.org/10.7554/eLife.40212.035

**Figure supplement 2.** Validation of PAR3 detection by antibody By western blot with anti-PAR3 antibody, we revealed numerous bands corresponding to several isoforms of PAR3 and to post-transcriptionally modified forms.
DOI: https://doi.org/10.7554/eLife.40212.036

way, we monitored only the PAR3 relocalisation process within the oocyte. We observed that after photobleaching, both by quantity and through normalised fluorescent measurement, PAR3 accumulates progressively at the anterior while being excluded from the posterior (*Figure 8B and C*). This indicates that PAR3, already present in oocytes, can be transported to the APM. The difference in quantity between the PAR3 which accumulates at the anterior and that which is excluded from the posterior, could be explained by the presence of PAR3 in transit into the cytoplasm.

We next addressed whether this transport in the oocyte relies on the MT cytoskeleton, as predicted from our results. We repeated PAR3 FRAP experiments in the presence of colchicin (*Figure 8D*). We observed that after photobleaching, the anterior accumulation and the posterior exclusion of PAR3 were significantly reduced (*Figure 8C*). Thus, this result shows that the MT network is important for anterior relocalisation of PAR3 fraction from the posterior in the oocyte.

## IKKε/IK2 controls asymmetrical localisation of PAR3

We have shown that PAR3 localisation depends on the MT network and its associated dynein motor. In the cytoplasm, PAR3 localised on RAB11-positive vesicles and RAB11 is important for PAR3 APM accumulation. Our results suggest that PAR3 is transported in association with MTs and recycling endosome vesicles to the APM. Interestingly, in the proteomic screen for SKTL partners, we also identified the IKKε/IK2 kinase. IKKε has been shown to regulate RAB11-associated cargo transport with dynein in developing bristles (*Otani et al., 2011*). Furthermore, IKKε phosphorylates a RAB11 cofactor, NUF, to release the cargo from the dynein motor at the MT minus end. Interestingly, in oocytes, IKKε is specifically enriched at the APM (*Dubin-Bar et al., 2008*). We then monitored whether IKKε is required for PAR3 polarised distribution. Upon IKKε RNAi-mediated knockdown in the oocyte, PAR3 becomes isotropic all along the plasma membrane, without clear accumulation at the APM (*Figure 9B*). Thereafter, PAR3 piles up in the cytoplasm (*Figure 9C*) in large circular structures surrounded by actin mesh that we called actin clumps (ACs) (*Figure 9A* and *Figure 9—figure supplement 1A*). We further noted that aPKC, a partner of PAR3, is also present on ACs (*Figure 9—figure supplement 1C*). Interestingly, ACs accumulate near MT minus ends close to the oocyte nucleus as revealed by Nod-LacZ transgene (*Clark et al., 1997*) (*Figure 9—figure supplement 2B–C*).

We then investigated the nature of those structures and their association with endocytic vesicles. ACs colocalise with RAB11 (*Figure 9D*) and NUF positive-vesicles (*Figure 9—figure supplement 1B*). RAB5 vesicles are not enriched in the ACs, but are all around them (*Figure 9E*). Furthermore, there is no colocalisation with Syntaxin16 highlighting the Golgi and lysosome compartment (*Akbar et al., 2009*) (*Figure 9—figure supplement 1D*).

These structures also contain SKTL (*Figure 9G*) and PI(4,5)P2 (*Figure 9F*), suggesting that PAR3, is still associated with PI(4,5)P2/SKTL membrane during its transport. ACs are strongly reduced when SKTL kinase dead form is expressed (SKTL$^{DNRQ}$, *Figure 9H*), and disappear completely in *sktl* mutant

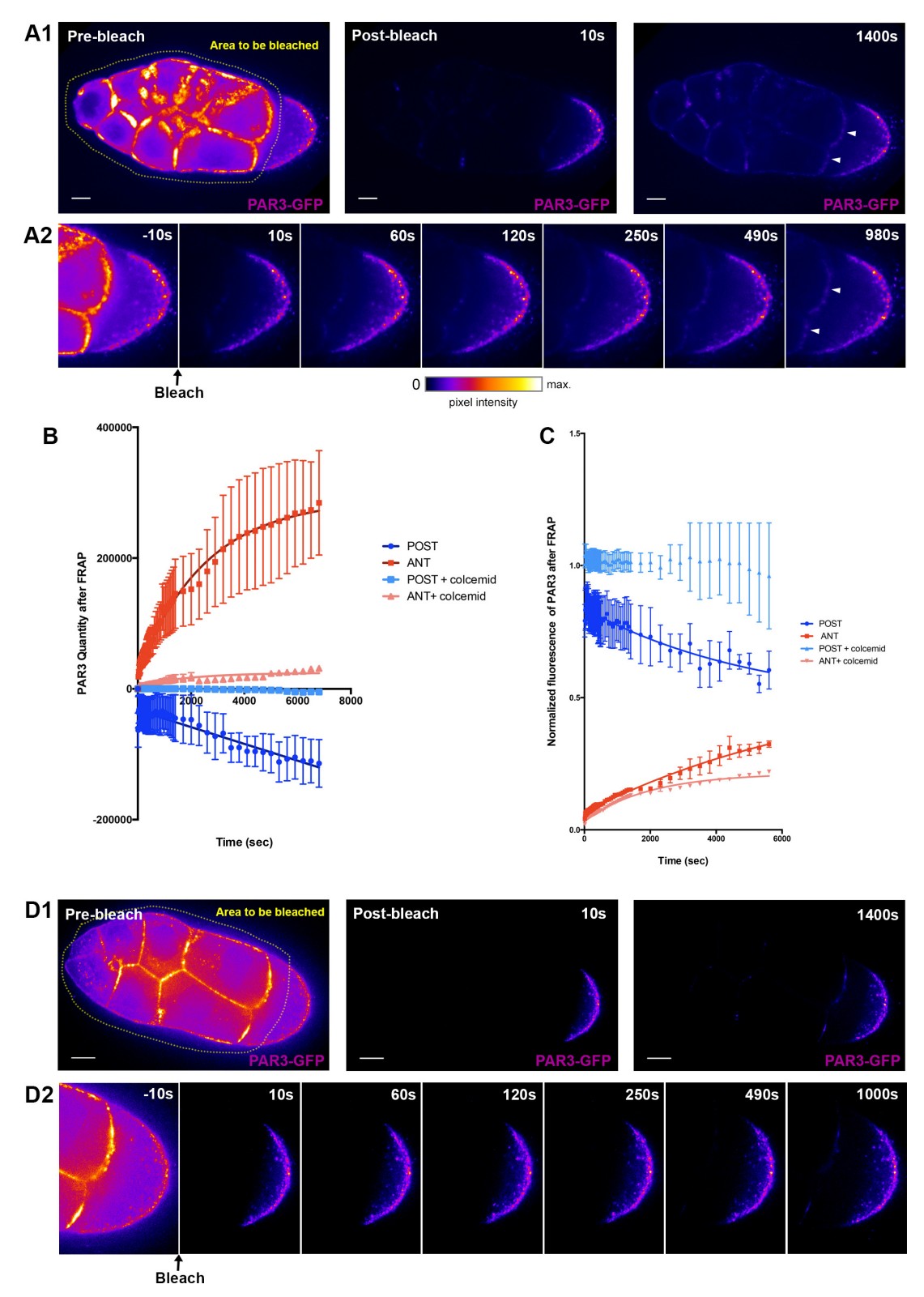

**Figure 8.** PAR3 recovery after APM photobleaching. (**A**) PAR3-GFP expressed in ovarian follicle (*Tub67c-GAL4; UASp PAR3-GFP*) is photobleached in all the nurse cells and at the APM (yellow area, **A1**). The fluorescence recovery was followed for around 1400 s (**A2**). (**B**) PAR3 quantity in each domain (anterior and posterior domains) was quantified using the same method as previously for three ovarian follicles and raised to 0 after bleaching. It can be seen that, after photobleaching, PAR3 accumulates progressively at the anterior while it is excluded from the posterior. (**C**) PAR3 quantity of each zone

*Figure 8 continued on next page*

*Figure 8 continued*

before FRAP was normalised to 1, and recovery of fluorescence was observed. (D) The same experiment as in (A) was performed on ovarian follicle incubated with colcemid. The quantification is shown in graphs in (C) and (D). In (B) and (C), the error bar represents SEM. The scale bars represent 20 µm.

DOI: https://doi.org/10.7554/eLife.40212.039

The following source data is available for figure 8:

**Source data 1.** PAR3 quantity variation (grey levels) at the anterior and the posterior of oocyte after anterior FRAP experiment.
DOI: https://doi.org/10.7554/eLife.40212.040

**Source data 2.** PAR3 quantity of each zone before FRAP was normalised to 1, and recovery of the fluorescence was observed.
DOI: https://doi.org/10.7554/eLife.40212.041

---

($sktl^{2.3}/sktl^{\Delta5}$, **Figure 9I**). Hence, in the absence of IKKε, SKTL activity and PI(4,5)P2 are necessary for AC accumulation.

Furthermore, in the absence of IKKε, the dynein subunit DLIC, which interacts with PAR3 and SKTL, is held back on structures similar to ACs (**Figure 9—figure supplement 2A**) at the MT minus ends (**Figure 9—figure supplement 2B–C**). Importantly, in the absence of MTs, upon colchicin treatment, ACs disappear, indicating that their accumulation depends on the MT network (**Figure 9K** compared to **Figure 9J**). Thus, in oocyte, IKKε seems to regulate the connection between PAR3 associated to RAB11-positive vesicles and MT minus ends.

## Discussion

Our results shed light on a dual step process that sustains PAR3 asymmetry in the *Drosophila* oocyte (**Figure 10**). The first step occurs at the middle of stage 9 and is responsible for PAR3 exclusion from the PPM. It involves PAR1, the actin cytoskeleton, PI(4,5)P2 and RAB5-dependent endocytosis. The second step relates to PAR3 accumulation at the oocyte anterior side. It brings into play MT-associated transport with the minus end directed motor, dynein in association with vesicular trafficking. Both steps rely on PAR3 interactions with PI(4,5)P2 endosomal vesicles.

The precise role of the actin cytoskeleton in exclusion of PAR3 from the PPM remains to be clarified. An attractive hypothesis is that the posterior actin network could act on PAR3 removal from the plasma membrane, possibly by direct regulation of the endocytosis process (**Figure 10C1**). In support of this, a connection between actin and a specific Oskar-dependent endocytosis pathway has already been highlighted at the oocyte posterior pole (**Tanaka and Nakamura, 2011**; **Vanzo and Ephrussi, 2002**; **Vazquez-Pianzola et al., 2014**). However, the actin requirement could also be connected with the role of actin in PAR1 posterior localisation (**Doerflinger et al., 2006**). Indeed, in the presence of latrunculin, PAR1 is not focused on the PPM but its localisation become isotropic. Consequently, posterior PAR3 would be less phosphorylated by PAR1 and so less excluded.

How is the posterior to anterior redistribution of PAR3 achieved in the oocyte? Recent analyses on MT-associated transport of mRNA particles in the oocyte have shown that the MT network is polarised with a slight bias for MT plus ends towards the posterior (**Trovisco et al., 2016**; **Zimyanin et al., 2008**). As PAR3 is associated with the MT minus end directed dynein motor,

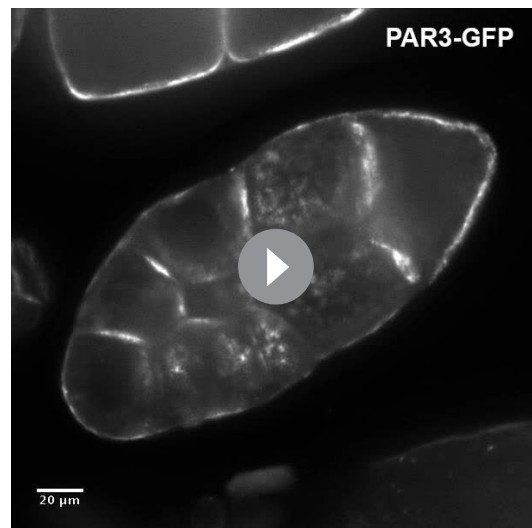

**Video 1.** Recovery of fluorescence after photobleaching of PAR3 at the APM domain. Representative FRAP experiment on PAR3-GFP (*Tub67c-GAL4; UASp PAR3-GFP*) in *Drosophila* egg chamber.
DOI: https://doi.org/10.7554/eLife.40212.042

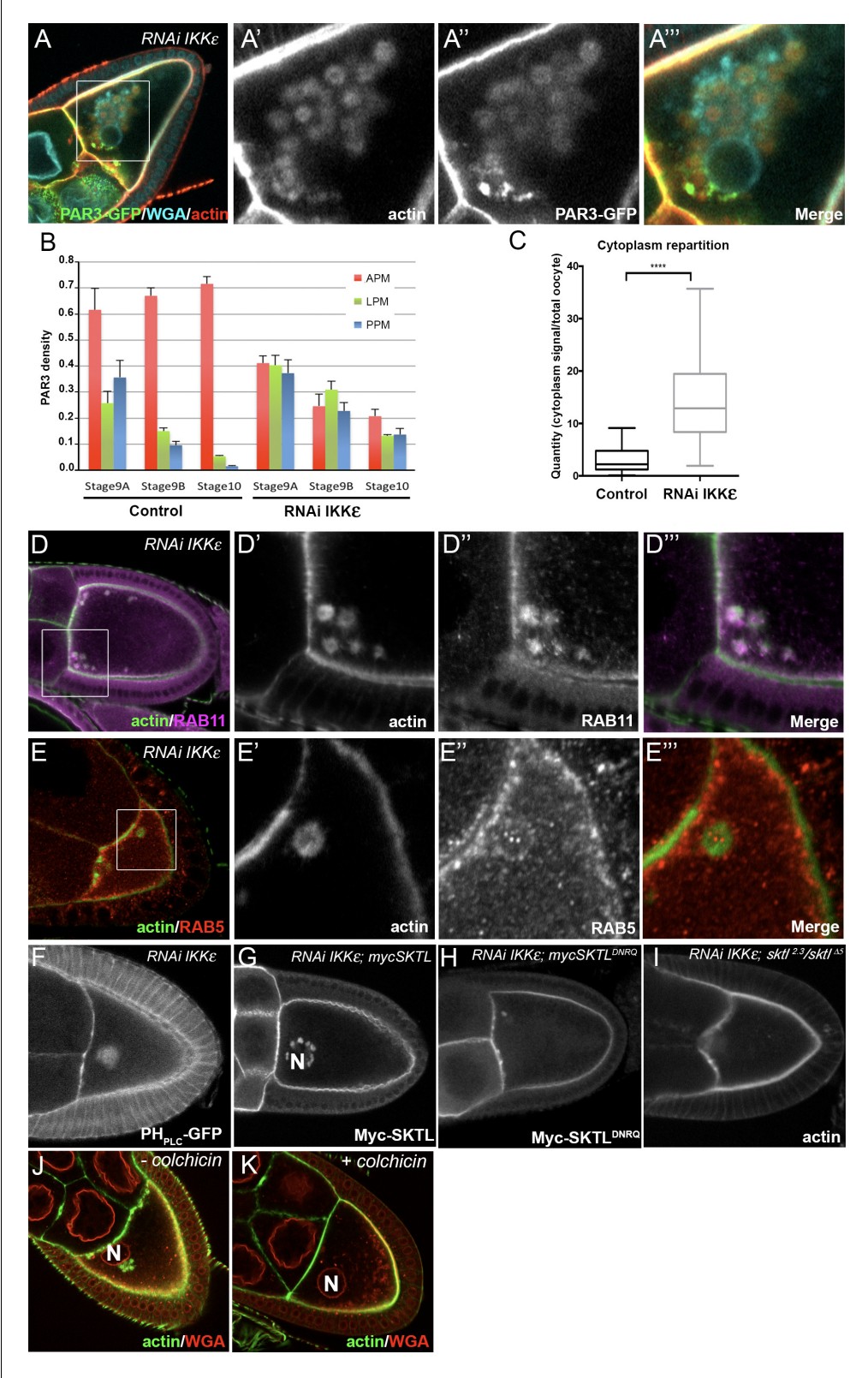

**Figure 9.** IKKε regulates PAR3 microtubule unloading and APM accumulation. (**A–C**) Distribution of PAR3 in response to IKKε knockdown. (**A**) IKKε knockdown affects the localisation of PAR3 and leads to an accumulation of circular actin clumps (ACs) enriched in PAR3. (**B**) Quantification of PAR3 density at each plasma membrane domain from stage 9A to stage 10 oocytes in WT or *IKKε* knockdown contexts (*Tub67c-GAL4; UASp RNAi ikkε; UASp PAR3-GFP*). (**C**) Quantification of PAR3-GFP distribution in the cytoplasm related to the whole oocyte intensity at stage 9B in WT or in *IKKε* knockdown

*Figure 9 continued on next page*

*Figure 9 continued*

contexts. For (**B, C**), Control (stage 9A, n = 9; stage 9B, n = 15; stage 10, n = 14); RNAi IKKε (stage 9A, n = 15; stage 9B, n = 10; stage 10, n = 9). (**C**) Mann-Whitney test. Error bars indicate SEM. **** indicates p<0.0001. (**D, E**) Actin clumps (**D', E'**, green) contain RAB11 (magenta, **D''**), but no RAB5 (**E''**, red) is around these actin clumps. Actin is visualised after staining with phalloidin. (**D'–D'''**) and (**E'–E'''**), are magnifications of **D** and **E** (white frame). (**F, G**) In an IKKε knockdown oocyte, ACs contain PI(4,5)P2, visualised with PH$_{PLC}$ GFP probe, (**F**) and Myc-SKTL, visualised with Myc tag (**G**). (**H, I**) PI(4,5)P2 or SKTL are involved in formation of the ACs. ACs are reduced in SKTL$^{DNRQ}$ context (**H**) and disappearin *sktl$^{2.3}$/sktl$^{Δ5}$* context (**I**). (**J, K**) MTs are necessary for actin ring formation. Flies were nourished with colchicin for 24 h (**K**) or only yeast paste (**J**). Actin (green) is visualised after staining with phalloidin and the nuclear membranes after staining with WGA (red). The oocyte nucleus position is indicated by an 'N' in (**G**), (**J**), and (**K**).

DOI: https://doi.org/10.7554/eLife.40212.043

The following source data and figure supplements are available for figure 9:

**Source data 1.** Quantification of PAR3 density at each plasma membrane domain in IKKe knockdown at stage 9B oocytes (Panel B).
DOI: https://doi.org/10.7554/eLife.40212.047
**Source data 2.** Quantification of PAR3 proportion in the cytoplasm related to the whole oocyte intensity in IKKe knockdown context.
DOI: https://doi.org/10.7554/eLife.40212.048
**Figure supplement 1.** Effect of IKKε knockdown.
DOI: https://doi.org/10.7554/eLife.40212.044
**Figure supplement 2.** Alteration of microtubule network in IKKε knockdown.
DOI: https://doi.org/10.7554/eLife.40212.045
**Figure supplement 2—source data 1.** Quantification of PAR3 in the cytoplasm related to the whole oocyte intensity in IKKe knockdown context.
DOI: https://doi.org/10.7554/eLife.40212.046

this MT polarity bias could be used to relocate PAR3 associated with vesicles towards the APM (*Figure 10A*). However, this anterior redistribution is unlikely to be direct because of the weak MT polarisation bias (*Khuc Trong et al., 2015*). Hence, it may involve intermediate re-localisation steps along the LPM before being redirected toward the APM, which is in accordance with the observed PAR3 distribution.

In the last few decades, the mutual inhibitions of PAR1 and PAR3 have been well studied at a functional level, and the molecular mechanism relies on some direct or indirect phosphorylations (*Figure 10C2*). However, the fate of PAR3 after phosphorylation by PAR1 at the posterior is unknown. PAR3 does not seem to be degraded in a SLMB/E3 ubiquitin ligase-dependent manner at the PPM, unlike aPKC or PAR6 (*Morais-de-Sá et al., 2014*). Using the FRAP approach, we have shown that posterior fractions of PAR3 proteins contribute after re-localisation through a MT-dependent process to the anterior pool of PAR3. It is important to mention that this anterior fraction of PAR3 coming from the posterior of the oocyte is likely to be minor compared to the anterior pool of PAR3, which comes directly from the nurse cells.

Our results indicate here that PAR3 and PAR1 coexist at the PPM until stage 9A, indicating that the relations between them both are more complex and that an additional process may be required to exclude PAR3 from the PPM. Starting from stage 9B, PAR1 excludes PAR3 from the PPM but has little effect on PAR3 anterior accumulation. However, the strict exclusion of PAR3 on this small posterior domain is sufficient to obtain an important anterior/posterior asymmetry.

PAR3 is a cytoplasmic protein, but here we show that it is often found in association with membranes, the plasma membrane or endocytic membrane (early/recycling endosome). Interestingly, in mammalian epithelial cells PAR3 associates with exocyst components (*Ahmed and Macara, 2017*). This association is present in the oocyte, but is also observed in nurse cells, particularly at the front of the ring canals where PAR3 accumulates in PI(4,5)P2 and RAB-positive compartments. How is PAR3, a cytosolic protein, transported with endocytic membranes? PAR3 could be associated to a cargo bound to vesicles, which would then be transported to the anterior. In parallel, PAR3 could also control its own association with the vesicles. Indeed, PAR3 interacts with SKTL, which, in return, produces PI(4,5)P2 that stabilises PAR3 to the membrane (*Claret et al., 2014*; *Krahn et al., 2010b*). This further points to the importance of SKTL in PAR3 APM accumulation and PPM exclusion. As PI(4,5)P2 is a well-known regulator of the first steps of endocytosis (*Posor et al., 2015*), we hypothesise that SKTL acts on PAR3 PPM exclusion by regulating the formation of vesicles required for future PAR3 anterior targeting. Our results also provide evidence for a role of PI(4,5)P2/SKTL in recycling endosome sorting. PI(4,5)P2 has been described as present both at the cell surface and on the distal portions of the tubular endosome (*Brown et al., 2001*). PI(4,5)P2 is required for the

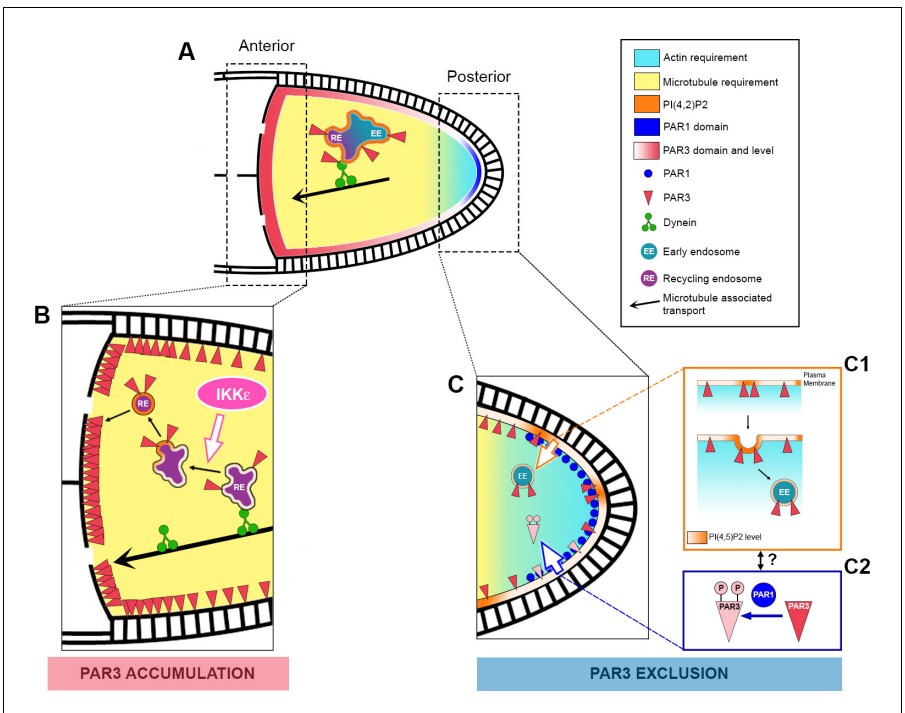

**Figure 10.** Speculative model of PAR3 localisation regulation. (**A**) In *Drosophila* oocytes, PAR3 asymmetrical localisation proceeds in at least two steps. The first step (**C**) occurs at the posterior plasma membrane and leads to the PAR3 exclusion starting from stage 9B. This step implicates PAR3 phosphorylation by PAR1 (**C2**) and RAB5-dependent endocytosis (**C1**). PIP5K, SKTL, and its product PI(4,5)P2 (orange) are crucial for this process, as well as the actin cytoskeleton (blue gradient). As PIP(4,5)P2 is critical for the endocytosis onset but also for actin cytoskeleton regulation, the link between all these protagonists remains hypothetical. Furthermore, as the polarised localisation of PAR1 depends on actin, we cannot exclude a role for PAR1 on endocytosis directly or indirectly by PAR3 phosphorylation. The second step (**B**) takes place at the anterior cortex where PAR3 has to be strongly enriched. Dynein-dependent transport brings PAR3 to the MT minus ends likely with recycling endosomal cargo (RE). Then PAR3 cargo would be released from the dynein through an IKKɛ-dependent process. However, how PAR3 reaches the cortex is unknown. PI(4,5)P2 is also important for the PAR3 endosomal sorting and we can speculate that an association with vesicles is required for anterior cortex targeting. Finally, while we have shown that the posterior fraction of PAR3 provides, through MTs, the anterior pool of PAR3, the neo-synthesized fraction of PAR3 from the nurse cell could also contribute to the anterior pool. As the MT network presents only a slight bias of minus end directed transport in the oocyte, it is thus possible that PAR3 is transported step by step along the lateral cortex before reaching the anterior membrane.

DOI: https://doi.org/10.7554/eLife.40212.049

recruitment to the membrane of many proteins involved in vesicle formation, fusion, and actin polymerisation (*Posor et al., 2015*). This suggests that PI(4,5)P2 and SKTL are important on membranes, including those of recycling endosomes, for the budding and the sorting of specific elements of PAR3.

Here, we show that IKKɛ participates in establishment of PAR3 asymmetry in the oocyte. Upon IKKɛ knockdown, PAR3 accumulates with aPKC on RAB11 positive-vesicle aggregates, suggesting that PAR3 transport at the MT minus ends is arrested. In polarised bristles, IKKɛ was previously shown to take the RAB11 endocytic vesicles down to the MT minus ends (*Otani et al., 2011*). The mechanism could be similar in oocytes (*Figure 10B*). In IKKɛ knockdown, vesicle aggregates accumulate at the MT minus ends with dynein subunit DLIC. This accumulation depends on MTs, but also on PIP(4,5)P2/SKTL. These vesicles enriched for PI(4,5)P2, SKTL and able to nucleate actin present common characteristics with the anterior plasma membrane. They could constitute a pre-built platform in the cytoplasm before being targeted to the appropriate position at the plasma membrane. This idea of a preformed platform has already been suggested in the tracheal system, where PAR3 associates with recycling endosomes containing E-cadherin during adherens junction rearrangements

(*Le Droguen et al., 2015*). Moreover, during de novo apical lumen formation in MDCK cells and in human pluripotent stem cells, there is an accumulation of apical components in RAB11 positive-vesicles that are subsequently delivered to the apical plasma membrane (*Overeem et al., 2015*; *Taniguchi et al., 2017*). The first sign of apical domain formation in these cells is the relocation of the polarity protein PAR3.

# Materials and methods

## Fiji macro and quantification methods

To quantify membrane and cytoplasm repartition of proteins in oocytes, we developed a macro on Fiji (cf. Oocyte Analysis macro). In each oocyte, we selected three points (two on both sides of the anterior membrane and one in the middle of the posterior membrane). Four sections of its plasma membrane were then automatically generated (anterior; lateral 1; lateral 2; posterior) independently of the oocyte stage (see *Figure 1—figure supplement 3* for details). After delimitation of the plasma membrane with the plot profile tool, we obtained its intensity profile. After delimitation of each plasma membrane domain with the polygon tool, we obtained the mean fluorescent intensity and the length of these domains. The anterior signal corresponds to the signal from the APM of the oocyte and the signal from the plasma membrane of the neighbouring nurse cells. Therefore, for each oocyte individually, we quantified the mean fluorescent intensity of a simple and a double plasma membrane of adjacent nurse cells. We then removed, from the anterior signal, the mean fluorescent intensity of the nurse cell plasma membrane to precisely quantify the signal coming only from the APM of this oocyte (*Figure 1-Figure supplement 3C*). Finally, after delimitation of the cytoplasm, we obtained its mean signal intensity. For each oocyte individually, the measured intensity signals were normalised in two steps. First, all the raw quantities (grey level intensity) were divided by the total intensity in the oocyte (APM + PPM + LPM + cytoplasm). Second, the plasma membrane values were expressed in density, that is the length of the corresponding membranes that divides them (see intermediate quantification in *Figure 1—figure supplement 2*). This process was performed on each oocyte individually, and then the results were pooled by genotypes and/or conditions and/or developmental stages. In our representation of the density, only the plasma membrane domains are indicated and not the cytoplasm. So the density of the different cortical zones does not add up to 1. The quantity corresponds to the grey level intensity measured on the image. The density represents the same values divided by the length of corresponding membranes. Between stage 9A and a stage 10, the oocyte plasma membrane overgrows. The density shows how PAR3 are concentrated at the membrane by avoiding the oocyte and plasma membrane size variation. The posterior exclusion ratio is the ratio of LPM density on PPM density. The asymmetry ratio is the ratio of APM density on PPM density.

To quantify the colocalisation of PAR3 with other proteins, the Coloc two plugin (Fiji) was used. Oocytes (between 7 and 13) of several females per genotype were analysed; the cytoplasmic contour was delimited and the quantification was performed only for one plan in this zone. The Pearson correlation coefficient (PC) or the Manders overlap coefficient (MOC) was determined in single confocal images. The PC value ranges from $-1$ (no correlation) to $+1$ (complete correlation), with values in between indicating different degrees of partial correlation. The negative control was provided by quantifying Pearson's coefficient for the same images, but after rotation of one by 90 degrees, a condition in which only random colocalisation is observed. The MOC measures the fraction of PAR3 signal overlapping with red signal and vice versa.

## Fly stocks

Mutant *sktl$^{l2.3}$* has been described in (*Gervais et al., 2008*), *sktl$^{\Delta5}$* in (*Hassan et al., 1998*), and *par-1$^{w3}$* and *par-1$^{6323}$* in (*Shulman et al., 2000*). The following fly stocks were also used: canton-S as wild type; *UASp-PAR3-GFP* and *UASp-PAR3$^{AA}$-GFP* (*Benton and St Johnston, 2003*); *UASp-GFP-PAR1 (N1S)* (*Doerflinger et al., 2006*); *Ubi-PH$_{PLC}$-RFP* (*Claret et al., 2014*); *pUbi-DLIC-GFP* (*Pandey et al., 2007*); *Pubi-PDI-GFP* (*Bobinnec et al., 2003*); *UASp-RAB5DN$^{(S43N)}$* (*Pelissier et al., 2003*); *FRT82B rab11$^{P2148/J2D1}$* (*Jankovics et al., 2001*); *Bac PAR3-GFP = P[w+, FRT9-2]18E, f, baz [815.8], P{CaryP, PB[BAC Baz-sfGFP2]attP18}* (*Besson et al., 2015*); the knockdown stocks from the Transgenic RNAi Project (TRiP, Bloomington Drosophila stock center) *UAS-RNAi par1$^{GL00253}$*; *UAS-*

RNAi dhc64$^{GL00543}$ (*Sanghavi et al., 2013*); *UAS-RNAi ikkε*$^{GL00160}$; *UAS-RNAi rab5*$^{HMS00147}$ (BL34832); *UAS-RNAi mCherry* (BL35785); PAR3-Trap = P{PTT-GC}baz$^{CC01941}$ (BL51572); *HspFLP; FRT82B RFP*; the stocks *UASp-Myc-SKTL* and *UASp-Myc-SKTL*$^{DNRQ}$, that we established. *Tub67c-GAL4* (*Januschke et al., 2002*) and *Osk-GAL4 VP16* (*Telley et al., 2012*) were used to express transgenes in germinal cells. *Tub67c-GAL4* was used to express all UAS transgenes, except the knockdown stock *UASp-RNAi dhc64*$^{GL00543}$ where *Osk-GAL4 VP16* was used. The crosses were kept at 25°C. However, to express two UAS transgenes, the flies were kept at 29°C, 24 h before dissection. *rab11*$^{P2148}$ germline clones were generated as described in *Compagnon et al. (2009)*, and selected against RFP.

## Drug treatment

Flies were fed with yeast paste, on vinegar agar plates, containing 1 mM of latrunculin B (Sigma), in DMSO, (sigma) or DMSO alone as control for 48 h, or 16 μM of colchicin (Sigma) for 24 h or 48 h. Ovaries were then dissected, fixed, and stained using standard procedures.

## Immunohistochemistry

Immunostainings were performed using standard protocols. The following primary antibodies were used: rabbit anti-SKTL at 1:20000 (*Claret et al., 2014*); mouse anti-Myc 9E10 (Santa Cruz) at 1:250; rabbit anti-RAB11 at 1:8000 (Nakamura); rabbit anti-RAB5 at 1:50 (Marcos Gonzalez-Gaitan); guinea pig anti-HRS at 1:500 (H. Bellen); mouse anti-KDEL at 1:300 (abcam 10C3, RRID: AB_298945); rabbit anti-NUF at 1:500; mouse anti-spNF at 1:50 (Abdu Uri); rabbit anti-Syntaxin16 at 1:1000 (R.Leborgne); rabbit anti-PKC (Santa Cruz) at 1:1000; mouse anti-HTS (Hybridoma, RRID: AB_528289) at 1:10; mouse anti-ß-Gal (Promega) at 1:250; rabbit anti-Staufen at 1:200. F-actin was visualised after staining with rhodamine-phalloidin (RRID: AB_2572408) or Alexa488-phalloidin (RRID: AB_2315147; life technology) at 1:200. Alexa594-WGA (Molecular probes) was used at 1:100 to stain nuclear membrane. Images were obtained with a ZEISS LSM700 confocal microscope and a Leica TCS-SP5 AOBS inverted scanning microscope.

## Molecular biology

SKTL kinase dead mutant, SKTL$^{DNRQ}$, was created using the PCR overlap mutagenesis method of insertion, as has been done on mammalian PIP5Kα (*Coppolino et al., 2002*), two punctual mutations in SKTL sequence (Asp$_{398}$ mutated in Asn and Arg$_{564}$ mutated in Gln).

Using the gateway recombination cloning, SKTL and SKTL$^{DNRQ}$ sequences were tagged with 6 MYC at the N terminal (in the vector pPMW, Murphy Lab). The sequence AttB used by the phiC31 integrase was inserted into the vector pPMW. The constructions were then integrated by transgenesis into the AttP2 site on chromosome III (BestGene).

## Co-immunoprecipitation and western blot

Ovary extracts were obtained from wild type and DLIC-GFP transgenic flies by dissecting ovaries (20 flies per genotype) into PBS. Ovaries were placed on ice with lysis buffer (10 mM Tris/Cl pH7.5; 150 mM NaCl; 0.5 mM EDTA; 0.5% NP-40 with Complete Protease Inhibitor cocktail, Roche) and mechanically homogenised using micro pestles in matching tubes. S2-(DGRC (RRID: CVCL_TZ72)) cell lysates were obtained after freeze-thaw lysis of cells transfected with GFP-SKTL with PAR3-HA. Ovaries and cells lysates were then spun at 16000 g for 10 min at 4°C. Using Chromotek standard procedures, the supernatant was collected and GFP-SKTL, PAR3-GFP, or DLIC-GFP were precipitated with 15 μL GFP-Trap_MA (Chromotek) magnetic beads for 2 h at 4°C with rotation. Input and bound fractions were analysed by SDS-PAGE and western blotting, using NuPage, 4–12% Bis-Tris gels (Life technology). DLIC was detected using 1:5000 guinea pig anti-DLIC C-ter (*Satoh et al., 2008*), PAR3-HA using rabbit 1:5000 Anti-HA (GenWay), PAR3 using 1:500 rabbit anti-PAR3 N-ter (*Wodarz et al., 1999*), PAR1 using 1:100 rabbit anti-PAR1 (Rb96, J. McDonald), GFP using 1:1000 mouse anti-GFP (Roche), or 1:5000 rabbit anti-GFP, Tubulin using 1:1000 mouse anti-tub (DM1A sigma). For the immunoprecipitation controls, we used PDI-GFP in fly extracts and GFP-SNAP (a kind gift from M. Sanial) in S2 cells extracts.

## Proteomic SKTL partner screen

S2 cells (S2-DGRC (RRID: CVCL_TZ72) were transfected with SKTL-GFP vector or not (control) and harvested 3 days later. Protein extraction and immunoprecipitation were performed using Chromotek standard procedures. After trypsin digestion, the samples were analysed by a LTQ Velos Orbitrap (Thermo Fisher Scientific). Among the potential partners identified, we focused on two proteins: DLIC (one peptide identified not present in the control: SGSPGTGGPGGAGNPAGPGR score 80) and IKKε (two peptides identified not present in the control: VMQQQQQEVMAVMR and LLAIEEDQEGR, score 24).

## FRAP experiment

Egg chambers were dissected in Schneider's Insect Medium supplemented with 13% FBS, 0.34 U/ml insulin. Egg chambers were cultured in a 150-µm-deep micro chamber filled with the same medium and sealed on one side with a 0.17 µm coverslip matching the characteristics of the facing objective lens, and a membrane permeable to oxygen on the other side. Imaging was carried out with spinning disk confocal (Axio Observer Z1 (Zeiss), Spinning Head CSU-X1, Camera sCMOS PRIME 95 (photometrics)) using a Plan-Apochromat 40×/1.25 oil objective lens. For each egg chamber, six prebleach images were taken. Then, the region of interest (ROI) corresponding to nurse cells and oocyte APM was scanned 25 times with the appropriate laser line at full laser power (473 nm) to photobleach GFP fluorescence. Fluorescence recovery into the ROI was monitored immediately after the bleach by time-lapse imaging at low-intensity illumination. Time-lapse recordings were processed with Fiji. The FRAP efficiency was controlled with the entire thickness of the sample.

The same experiment was realized on egg chambers treated with colcemid. The egg chambers were incubated for 30 min in Schneider medium, 2% FBS, 0.2 U/ml insulin with 0.82 µM of colcemid, before the onset of the FRAP experiment, and the egg chambers stayed in this medium during the fluorescence recovery.

## Endocytosis assay

To confirm the effect of RAB5DN expression on endocytosis, we performed an endocytosis assay using the lipophilic dye FM4-64 (Molecular probes). This dye partitions into membranes where it becomes fluorescent and can be internalised in the cell by endocytosis.

Egg chambers were dissected in Schneider's Insect Medium supplemented with 20% FBS, 10 µg/ml of juvenile hormone, and 0.2 U/ml insulin. Next, they were incubated in the dark with the same medium +20 µM FM4-64 for 45 min at 25°C and then rinsed two times and washed for 15 min at 4°C in medium without FM4-64. The oocytes were immediately observed by confocal microscopy.

## Statistical analysis

Data were analysed using the Mann-Whitney test and were represented by box plot (with Tukey variation) thanks to Graphpad Prism 6. All plots are expressed as the mean ± standard error of the mean (SEM).

## Acknowledgements

We acknowledge the ImagoSeine core facility of the Institut Jacques Monod, member of IBiSA and the France-BioImaging (ANR-10-INBS-04) infrastructure. We thank the proteomic platform of Institut Jacques Monod. We thank A Wodarz, J McDonald, D St Johnston, R Karess, F Schweisguth, M Sanial, and M Erdélyi for reagents. We thank F Bernard, C Jackson, JA Lepesant, and L Pintard for critical comments on the manuscript; C Seiler for technical assistance; L Gervais for directed mutagenesis of SKTL[DNRQ]; the Bloomington Stock Center for fly stocks; the Developmental Studies Hybridoma Bank for monoclonal antibodies; BestGene Inc Service for *Drosophila* embryo injections. This work was supported by the CNRS, by the ARC (grant SLR20130607102, grant PJA 20141201756 and PJA 20161204931), and by the 'Ligue Contre le Cancer' (grant RS14/75-58).

JJ was supported by a fellowship from the Ministry of Research and Technology (MRT) and by a fellowship from Fondation 'Ligue Contre le Cancer'. The authors declare that they have no financial interest that might influence the results or interpretation of their manuscript.

# Additional information

## Funding

| Funder | Grant reference number | Author |
|---|---|---|
| Ministère de l'Enseignement Supérieur, de la Recherche Scientifique | | Julie Jouette |
| Ligue Contre le Cancer | | Julie Jouette |
| Fondation ARC pour la Recherche sur le Cancer | SLR20130607102 | Antoine Guichet |
| Ligue Contre le Cancer | RS14/75-58 | Antoine Guichet |
| Fondation ARC pour la Recherche sur le Cancer | PJA 20141201756 | Antoine Guichet |
| Fondation ARC pour la Recherche sur le Cancer | PJA 20161204931 | Antoine Guichet |

The funders had no role in study design, data collection and interpretation, or the decision to submit the work for publication.

## Author contributions

Julie Jouette, Conceptualization, Data curation, Formal analysis, Investigation, Visualization, Methodology, Writing—original draft, Performed experiments and data analysis, Conducted data interpretation and writing; Antoine Guichet, Funding acquisition, Project administration, Writing—review and editing, Conducted data interpretation and writing; Sandra B Claret, Conceptualization, Data curation, Software, Formal analysis, Supervision, Funding acquisition, Validation, Investigation, Visualization, Methodology, Writing—original draft, Project administration, Writing—review and editing, Performed experiments and data analysis, Conducted data interpretation and writing

## Author ORCIDs

Julie Jouette ⓘ http://orcid.org/0000-0002-0115-0065
Antoine Guichet ⓘ http://orcid.org/0000-0001-7216-1944
Sandra B Claret ⓘ http://orcid.org/0000-0001-7167-510X

## Decision letter and Author response

Decision letter https://doi.org/10.7554/eLife.40212.056
Author response https://doi.org/10.7554/eLife.40212.057

# Additional files

## Supplementary files

• Source code 1. Oocyte analysis source code.
DOI: https://doi.org/10.7554/eLife.40212.050

• Transparent reporting form
DOI: https://doi.org/10.7554/eLife.40212.051

## Data availability

All data generated or analysed during this study are included in the manuscript and supporting files.

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
