## [Decision Letter]

[Editors’ note: a previous version of this study was rejected after peer review, but the authors submitted for reconsideration. The first decision letter after peer review is shown below.]

Thank you for submitting your work entitled "Dynein-mediated transport and membrane trafficking control PAR3 polarised distribution" for consideration by *eLife*. Your article has been reviewed by a Senior Editor, a Reviewing Editor, and three reviewers. The reviewers have opted to remain anonymous.

Our decision has been reached after consultation between the reviewers. Based on these discussions and the individual reviews below, we regret to inform you that your work will not be considered further for publication in *eLife*.

You will see from the individual reviews below that there was significant appreciation of the potential interest of this work. Overall, however, the reviewers felt that there were too many significant revisions that would need to be done. There was, however, a feeling that this work might become appropriate for *eLife* were it to be further developed, and the reviewers did not preclude future submission of a more mature story. It would, however, be treated as a new submission.

*Reviewer #1:*

In this manuscript, Jouette and colleagues examine the mechanism by which opposing gradients of Par1 and Par3 are established in the *Drosophila* oocyte. These polarity proteins, and their mutual antagonism are critical to polarity establishment in a number of cell types and a variety of organisms. This is a highly significant topic and an understanding of this mechanism will be of interest to a wide audience.

The main focus of this paper is the mechanism by which Par3 is enriched along the anterior cortex of the oocyte and excluded from the posterior pole. Initial studies suggested that posterior exclusion was predominantly via Par1-mediated phosphorylation of Par3. The authors convincingly show that although this mechanism is important in stage 10 egg chambers, additional factors are also at play. The authors further suggest that Dynein, microtubules and endocytic trafficking are responsible for targeting Par3 to the anterior cortex. This latter point, however, is not convincingly supported by the data presented. In general, there are some very interesting findings in this paper. However, there are also major deficits in interpretation of results and in the controls that are used. Ideally, these issues would be fixed prior to publication.

Essential revisions:

The authors conclude that Par3 interacts with Dlic. The interaction is very hard to appreciate from the data that is presented. The enrichment of the Par3 bands in the "+" lane is barely above background. Secondly, why are there four bands corresponding to Par3? Are these degradation products or splice isoforms? According to FlyBase, the four isoforms of Par3 have sizes of 157, 163, 161 and 159 kD. That is not what we see here. The most prominent band appears to be around 55kD. The Par3-HA band that is shown in Figure 4G is around 250kD.

Figure 3. Par3-GFP is found to accumulate in small vesicles that are positive for π (4,5)P2. This appears convincing. However, the authors go on to state that these same vesicles are also positive for Rab5 and Rab11. The localization of Rab5 and Rab11 to these vesicles is not nearly as convincing. The authors have done a good job of quantifying the localization of Par3 along the membrane. It would be good to also quantify the co-localization of Rab5 and Rab11 with these Par3-GFP vesicles (for instance using a Pearson's coefficient analysis).

The connection between SKTL activity and Par3 localization is convincing. The authors then go on to state that protein interaction partners of SKTL were identified using purification and mass spectrometry. However, the data for this mass spec experiment is not shown. How many peptides of Par3 were found in the SKTL pellet in comparison to the control? The authors then go on to validate this interaction. For this experiment, they used S2 cells. What was the rationale for using S2 cells? All of the rest of the experiments were performed in the ovary. Does this interaction occur in ovarian lysates? An additional point about experimental setup. The negative control for this experiment appears to be "no antibody". A better negative control would be to use cells or extracts expressing GFP alone.

The authors conclude that IKKε regulates transport of recycling endosomes in the oocyte. Although this is possible, the data that is presented do not provide proof of this. What is shown is that depletion of IKKε results in abnormal actin structures at the anterior of the oocyte that also contain Rab11. The images presented are from fixed samples that are stained with antibody. The authors suggest that these vesicles are formed due to blocking of an unloading step. While this is certainly possible, the data only addresses localization.

The authors suggest that when the anterior (high)-posterior (low) distribution of Par3 is altered, oocyte polarity is affected and the oocyte nucleus is mis-localized. However, from Figure 2 it appears that Latrunculin treatment affects this Par3 AP distribution to a greater extent than Colchicin treatment. Yet, the oocyte nucleus is localized under Latrunculin conditions but delocalized under Colchicin treatment conditions. This does not seem consistent.

*Reviewer #2:*

In this study, Julie and coworkers quantify the plasma membrane distribution of Par3 in the *Drosophila* oocyte as it matures. During this process, Par3 becomes excluded from the posterior (PPM) and accumulates at the anterior plasma membrane (APM). In addition to the well-known effects of Par1, the authors report that exclusion from the PPM depends on Rab5 and PIP2-dependent endocytosis. They also argue that microtubules and dynein are required for transport to the APM, where it is unloaded through an IKK-related kinase. Interestingly, they show that Par3 can bind to Sktl, a PI4P5-kinase.

This is an interesting study and the careful quantification of Par3 localization is a useful approach. However, I found the work overall to be rather diffuse and the interpretations were highly over-stated. At a general level, it is not clear to me how the authors know that Par3 is in fact being transported from the PPM to the APM – each event (loss from the PPM and accumulation at the APM) could be independent of one another, and this possibility is not excluded in this study. Indeed, there does not seem to be any net gain of Par3 at the APM from stages 8 – 10 (Figure 1). Par3 association with membranes is highly dynamic, through a process likely controlled by phospho-inositide binding, oligomerization, and phosphorylation. It might diffuse through the cytoplasm and simply bind to any membrane with high levels of PIP2. In addition, new Par3 protein could be delivered from the nurse cells.

Essential revisions:

1) The studies of the role of actin and microtubules are rather superficial – for instance the actin data are based only on latrunculin treatment, and the authors assume, based only on the similar phenotype induced by loss of Par1 or expression of a non-phosphorylatable Par3, that the actin network is mediating the effects of Par1 on Par3. This is not a valid conclusion – the same phenotype could be generated through 2 independent mechanisms.

2) Figure 2J (co-IP of Par3 with DLIC-GFP) is not convincing. The negative control lane input is lower than the positive. Also, a better negative control is needed (another GFP fusion protein); the DLIC-GFP input needs to be shown; and some evidence that the indicated bands actually represent Par3 needs to be provided. And in Figure 2—figure supplement 2B an input control is needed.

3) Figure 3. The authors claim that Par3 particles are associated with PIP2-containing membranes and that Par3 is removed from the PPM by endocytosis on Rab5/PIP2 endosomes, but in Figure 3, most of the Par3 in cytoplasmic spots is NOT associated with Rab5 OR PIP2; and the single Rab5+ vesicle is much larger than any of the PIP2 positive spots. Overall, these data do not agree with the stated hypothesis. The staining for Rab5 and Rab11 is also very diffuse – it would be better to use FP-tagged Rabs for this type of experiment.

4) In Figure 3—figure supplement 1, how do we know the antibodies are specific? – the HRS antibody does not seem to detect vesicles.; and the syntaxin16 antibody decorates the plasma membrane but no detectable trans-Golgi.

5) The interaction between SKTL and Par3 is based only on over-expressed proteins. The blot needs the GFP-SKTl input and a better (GFP fusion) negative control. Also, I could not find data to support the contention that Par3 forms a tripartite complex with SKTL and PIP2, as is stated in subsection “SKTL and PAR3 act in synergy to form vesicular platforms”. The authors also suggest that Par3 could regulate SKTL activity, and that this would provide a positive feedback loop, but again without any supporting evidence.

6) In Figure 5 it looks as though the mutant SKTL is expressed at lower levels than the WT. Could this, rather than loss of kinase activity, explain the lower GFP-Par3 in cytoplasmic spots? The colocalization with Rab5 and 11 is not convincing. Moreover, it looks as though the aAPM has disintegrated – is this not an issue with interpretation of the data?

7) Finally, I think it is premature to conclude much from the IKK-eta data – clearly knockdown generates some kind of cytoplasmic inclusion in the oocytes, that accumulate actin and Rab11, but it’s not clear what they are; and since Par3-GFP is still strongly accumulated all over the plasma membrane, including the APM (Figure 6A – I), I don't think that it is clear that IKKe is necessary for unloading it from microtubules.

8) In general, the authors seem to interpret their data as if Par3 is an integral membrane protein that is being endocytosed and transported by the microtubule machinery, but of course it is not – it is a peripheral protein that associates with negatively-charged phospholipids in a highly dynamic way and is presumably able to diffuse through the cytoplasm between vesicles and plasma membrane.

*Reviewer #3:*

Jouette and colleagues have examined mechanisms that position the polarity protein Baz/Par-3 in a well-established model of cell polarization, the *Drosophila* oocyte. They examine a number of mechanisms involved in the cortical removal, sub-cellular trafficking and intracellular compartment release of Baz. The trafficking of Baz through the cytosol is poorly understood, and the results of this paper make significant advances in elucidating this process. However, major concerns should be addressed.

Major concerns

1) All of the Baz distributions are determined with an over-expressed Baz construct. Thus, it is unclear how relevant identified mechanisms are for endogenous levels of the protein. The newly identified mechanisms should be additionally evaluated with probes for endogenous Baz (using either Baz antibodies or an available Baz-GFP gene trap line).

2) Connections between the mechanisms that remove Baz from the cortex are unclear. Possible connections should be investigated with the tools already used to determine if the mechanisms are independent or interdependent.

2.1) Abnormal Baz accumulations arise along the later cortex with dynein RNAi. Are these accumulations dependent on cortical removal steps mediated by Par-1, PIP2 or Rab5? Perturbations of these proteins in pair-wise combinations could reveal dependent or independent cortical removal mechanisms.

2.2) Skittles overexpression is shown to increase the removal of Baz from the posterior domain. Does this removal require Par-1 or Rab-5? Again, combining the Skittles overexpression with the Par-1 RNAi or the Rab5DN could address how interdependent the removal mechanisms are.

3) Since Baz becomes isotropic around the cortex with IKK-epsilon knock down, does IKK-epsilon also play a role in the removal of Baz from the posterior cortex? If IKK-epsilon only promoted delivery to the anterior cortex, then removal from the posterior would still be expected. Some assessment of IKK-epsilon's direct or indirect role at the posterior would be expected within the overall context of the paper.

[Editors’ note: what now follows is the decision letter after the authors submitted for further consideration.]

Thank you for submitting your article "Dynein-mediated transport and membrane trafficking control PAR3 polarised distribution" for consideration by *eLife*. Your article has been reviewed by Anna Akhmanova as the Senior Editor, a Reviewing Editor, and two reviewers. The reviewers have opted to remain anonymous.

The reviewers have discussed the reviews with one another and the Reviewing Editor has drafted this decision to help you prepare a revised submission.

This manuscript (an earlier version of which which has been previously reviewed by *eLife* but is now being treated as a new submission rather than a resubmission) provides evidence for a pathway that polarizes the distribution of Baz (Par-3) across the *Drosophila* oocyte, an important model of cell polarity. The model is an interesting and, if fully supported by the data, would be appropriate for publication in *eLife*. However, there are significant concerns that would need to be addressed before it could be accepted.

Essential revisions:

1) Generally, the quality of the biochemical evidence is not as strong as it should be. For example, in Figure 7E and F, the IPs of GFP-SKTL testing for pull-down of PAR3 and DLIC, the control was no addition of GFP antibody. Since addition of antibodies can increase non-specific binding, the better control is to IP an irrelevant GFP protein with the GFP antibody (as done in Figure 7D). Especially when the protein interaction they are trying to show is weak, good controls are required.

Even in 7D, which does have an appropriate control, the co-IP is pretty weak. This does not invalidate it, but it should be acknowledged in the text.

In the earlier version it was unclear that the 250kD band that was observed on western blots actually corresponded to Par3. The authors have used RNAi strains to address this point and the conclusion is a bit more convincing. It is still hard to fully appreciate, though. For instance, in Figure 7—figure supplement 2, a strain over-expressing Par3-GFP using the UAS-Gal4 driver is used. These embryos should be expressing a relatively high level of Par-GFP. In the total fraction from these embryos the most prominent band is around 100kD when probed with a Par3 antibody. However, this band is hard to detect with the GFP antibody. In fact, it is hard to see any bands in the GFP blot. Next, the sample was immunoprecipitated. It is unclear from the legend which antibody (Par3 or GFP) was used in the IP. In the immunoprecipitated fractions, the most prominent band is the 250kD band. The band is reduced in the strain expressing Par3 RNAi but it is not clear how it can be detected so prominently in the IP fraction when the same isoform is a minor species at best in the total fraction. Is the IP antibody somehow specifically precipitating an oligomeric form? The experiment with the endogenously expressed BAC-based Par3-GFP construct is more convincing.

2) Another major issue is the over-reliance on overexpressed Par3. Key results should be repeated to look at endogenous Par3 expression. For example, are the Par3-positive cytoplasmic particles, and effects on these particles, observed when Par3 is expressed by its own regulatory regions? Without this additional data, there is a risk that the effects they report are artifacts of overexpression.

3) The results using a dominant negative rab5 should be confirmed with an shRNA construct (available from Bloomington). There is a risk that the DN might produce off target effects.

---

## [Author Response]

[Editors’ note: the author responses to the first round of peer review follow.]

Our decision has been reached after consultation between the reviewers. Based on these discussions and the individual reviews below, we regret to inform you that your work will not be considered further for publication in eLife.You will see from the individual reviews below that there was significant appreciation of the potential interest of this work. Overall, however, the reviewers felt that there were too many significant revisions that would need to be done. There was, however, a feeling that this work might become appropriate for eLife were it to be further developed, and the reviewers did not preclude future submission of a more mature story. It would, however, be treated as a new submission.

Reviewer #1:

In this manuscript, Jouette and colleagues examine the mechanism by which opposing gradients of Par1 and Par3 are established in the Drosophila oocyte. These polarity proteins, and their mutual antagonism are critical to polarity establishment in a number of cell types and a variety of organisms. This is a highly significant topic and an understanding of this mechanism will be of interest to a wide audience.The main focus of this paper is the mechanism by which Par3 is enriched along the anterior cortex of the oocyte and excluded from the posterior pole. Initial studies suggested that posterior exclusion was predominantly via Par1-mediated phosphorylation of Par3. The authors convincingly show that although this mechanism is important in stage 10 egg chambers, additional factors are also at play. The authors further suggest that Dynein, microtubules and endocytic trafficking are responsible for targeting Par3 to the anterior cortex. This latter point, however, is not convincingly supported by the data presented. In general, there are some very interesting findings in this paper. However, there are also major deficits in interpretation of results and in the controls that are used. Ideally, these issues would be fixed prior to publication.Essential revisions:1) The authors conclude that Par3 interacts with Dlic. The interaction is very hard to appreciate from the data that is presented. The enrichment of the Par3 bands in the "+" lane is barely above background. Secondly, why are there four bands corresponding to Par3? Are these degradation products or splice isoforms? According to FlyBase, the four isoforms of Par3 have sizes of 157, 163, 161 and 159 kD. That is not what we see here. The most prominent band appears to be around 55kD. The Par3-HA band that is shown in Figure 4G is around 250kD.

As suggested by the reviewer, we have reiterated several times the IP experience of DLICGFP to validate the interaction between DLIC and PAR3. We provide a new blot presented in the Figure 7. In the course of those new experiments, as our previous PAR3 antibody made in rat was ran out, we have used a different PAR3 antibody produced in rabbit and kindly provided by A. Wodarz. We still observed several bands between 250 and 115 kDA. We can see that one of these bands (250kDa) immunoprecipitates with DLIC and this immunoprecipitation is reproducible. We have verified that this band corresponds to PAR3 and not to a non-specific protein. For this extend, we used a *Drosophila* line that produces PAR3-GFP at an endogenous level, and we revealed, after western blot on ovarian extracts, PAR3 with PAR3 antibody or with a GFP antibody. To better visualize PAR3, we have also realised an IP with the same extracts. The >250 kDa band is detected by PAR3 and GFP antibodies, indicating that the 250kDa band is really PAR3 (Figure 7—figure supplement 2). Furthermore, we have also validated the specificity of PAR3 antibody by monitoring PAR3 knockdown after RNAi. (Figure 7—figure supplement 2). The abnormal weight of PAR3 could be explained by the presence of a dimeric form of PAR3, the oligomerisation of PAR3 being already described (Benton and St Johnston, 2003; Dickinson et al., 2017; Kullmann and Krahn, 2018). Moreover, high molecular weight bands are also shown in several papers without explanations (Krahn et al., 2010; Krahn et al., 2009).

2) Figure 3. Par3-GFP is found to accumulate in small vesicles that are positive for π (4,5)P2. This appears convincing. However, the authors go on to state that these same vesicles are also positive for Rab5 and Rab11. The localization of Rab5 and Rab11 to these vesicles is not nearly as convincing. The authors have done a good job of quantifying the localization of Par3 along the membrane. It would be good to also quantify the co-localization of Rab5 and Rab11 with these Par3-GFP vesicles (for instance using a Pearson's coefficient analysis).

As suggested by the reviewer, we have quantified the colocalisation of PAR3 with PI(4,5)P2, RAB5 and RAB11 vesicles. The Pearson’s coefficient was obtained by using the FiJi Macro Coloc2 on our images. The negative control was provided by quantifying colocalisation for the same images, but after rotation of one by 90 degrees, a condition in which only random colocalisation is observed. The results are now presented in the manuscript Figure 5A. As indicated in the text, the PAR3 particles do not colocalise with the endoplasmic reticulum (KDEL), the Golgi apparatus (SYX16) or late endosomes (HRS). However, they colocalise with PI(4,5)P2 and RAB5 enriched compartments and to a lesser extent with RAB11 enriched compartments.

3) The connection between SKTL activity and Par3 localization is convincing. The authors then go on to state that protein interaction partners of SKTL were identified using purification and mass spectrometry. However, the data for this mass spec experiment is not shown. How many peptides of Par3 were found in the SKTL pellet in comparison to the control? The authors then go on to validate this interaction. For this experiment, they used S2 cells. What was the rationale for using S2 cells? All of the rest of the experiments were performed in the ovary. Does this interaction occur in ovarian lysates? An additional point about experimental setup. The negative control for this experiment appears to be "no antibody". A better negative control would be to use cells or extracts expressing GFP alone.

We feel that the reviewer has misinterpreted our results. We have performed a proteomic screen to identify some SKTL partners but we do not found PAR3. We have found DLIC and IKKε amongst others. However, as suggested by the reviewer, we have added for DLIC and IKKε the mass spec data in the Materials and methods section.

“DLIC: SGSPGTGGPGGAGNPAGPGR score 80.

IKKε: VMQQQQQEVMAVMR and LLAIEEDQEGR score 24.

The mass spectrometry experiment has been performed by using S2 cells extract. We agree with the reviewer that this experiment would be more informative if we had been able to use ovarian extract. The expression of GFP-SKTL is particularly deleterious in the *Drosophila* oocyte and we cannot perform the immunoprecipitation (IP) with SKTL overexpressed extract. We tried to immunoprecipitate a GFP-SKTL form produced at endogenous level in a CRISPR transgenic line without success. Moreover, the SKTL antibody is not good enough to use it. Therefore, we used S2 cells to realize this mass spectrometry experiment. However, to validate the interaction between PAR3 and DLIC, we performed the IP with ovarian extracts.

These peptides are not present in control.”

Concerning the negative control of this experiment, we have realized two different types of control. As suggested by the reviewer, we have tested the IP with another GFP transgene (PDI-GFP) and have observed no co-IP of PAR3 or DLIC. But we have also performed the co-IP on the same extract with or without anti-GFP antibody. We had chosen to show this last control because if we overexpressed one of these proteins, we can modify the migration profile of others. For example, PAR3 expression modifies the migration profile of DLIC and conversely (variation of postraductional modifications or change in isoforms ratio). We feel that it can be hazardous to compare two wells in which the protein contents are very different. In the case of DLIC/PAR3 IP (Figure 7D), we show now the IP of DLIC-GFP and in control the IP of PDI-GFP.

4) The authors conclude that IKKε regulates transport of recycling endosomes in the oocyte. Although this is possible, the data that is presented do not provide proof of this. What is shown is that depletion of IKKε results in abnormal actin structures at the anterior of the oocyte that also contain Rab11. The images presented are from fixed samples that are stained with antibody. The authors suggest that these vesicles are formed due to blocking of an unloading step. While this is certainly possible, the data only addresses localization.

We agree with the reviewer that this experiment is indicative, but is not sufficient to support a strong conclusion. Our conclusions on recycling endosome transport are built upon confocal analysis on fixed issues and not through live imaging analysis. The published data concerning IKKε functions in *Drosophila* bristles report that IKKε regulates the localisation of RAB11 and Dynein along with the rapid shuttling of recycling endosomes (Otani et al., 2015; Otani et al., 2011). The activation of IKKε by a local phosphorylation and, the spNF interaction with IKKε and dynein are conserved between bristle and oocyte in *Drosophila* (Amsalem et al., 2013; Otani et al., 2015). With our results, we can conclude that microtubules are important to the accumulation of RAB11 positive-vesicles near the microtubules minus ends as it has been reported in the *Drosophila* bristles. But we cannot conclude to a direct transport of endosomes on microtubules; we can only indicate that our, like the others suggest this point. So as requested by the reviewer we have modified this point in the Results section of the new version of the manuscript.

5) The authors suggest that when the anterior (high)-posterior (low) distribution of Par3 is altered, oocyte polarity is affected and the oocyte nucleus is mis-localized. However, from Figure 2 it appears that Latrunculin treatment affects this Par3 AP distribution to a greater extent than Colchicin treatment. Yet, the oocyte nucleus is localized under Latrunculin conditions but delocalized under Colchicin treatment conditions. This does not seem consistent.

The reviewer raises an important point. Indeed, what seems important to maintain a correct polarity within the oocyte (with the nucleus positioning as readout), is the PAR3 asymmetry between the anterior and posterior rather than the posterior exclusion of PAR3 stricto senso. For example, in presence of latrunculin, the posterior exclusion of PAR3 is affected but its anterior accumulation a very little. Thus, the ratio between anterior and posterior remains high, and the nucleus is well located. However, in presence of colchicin, the posterior exclusion is always observable but there is no/weak anterior accumulation. So, the ant/post ratio is weak and the nucleus is mispositionned. In the Figure 2, the ant/post ratio (asymmetry ratio) is not indicated and we propose to add this to Figure 2 in the revised manuscript.

Reviewer #2:

In this study, Julie and coworkers quantify the plasma membrane distribution of Par3 in the Drosophila oocyte as it matures. During this process, Par3 becomes excluded from the posterior (PPM) and accumulates at the anterior plasma membrane (APM). In addition to the well-known effects of Par1, the authors report that exclusion from the PPM depends on Rab5 and PIP2-dependent endocytosis. They also argue that microtubules and dynein are required for transport to the APM, where it is unloaded through an IKK-related kinase. Interestingly, they show that Par3 can bind to Sktl, a PI4P5-kinase.This is an interesting study and the careful quantification of Par3 localization is a useful approach. However, I found the work overall to be rather diffuse and the interpretations were highly over-stated. At a general level, it is not clear to me how the authors know that Par3 is in fact being transported from the PPM to the APM – each event (loss from the PPM and accumulation at the APM) could be independent of one another, and this possibility is not excluded in this study. Indeed, there does not seem to be any net gain of Par3 at the APM from stages 8 – 10 (Figure 1). Par3 association with membranes is highly dynamic, through a process likely controlled by phospho-inositide binding, oligomerization, and phosphorylation. It might diffuse through the cytoplasm and simply bind to any membrane with high levels of PIP2. In addition, new Par3 protein could be delivered from the nurse cells.

The reviewer raises two very important points concerning PAR3 localisation mechanism.

The first point concerns the posterior PAR3 accumulation and the origin of its accumulation. Currently we present the PAR3 quantification, normalised to the total amount of signal in the oocyte and related to the membrane length (density). This representation avoids some fluctuations associated to the membrane growth during oogenesis or to experimental procedures but also allows us to compare different genotypes. Thus, in our graphs, we can follow the evolution of asymmetry, but we cannot compare the quantity between different stages. To respond to the reviewer, we have added a new graph with the raw data of PAR3 quantity (grey levels) in oocyte (Figure 1D). We have also presented in sup data an intermediate step of the quantification in which PAR3 density are shown without normalisation (Figure 1—figure supplement 2).

So, we can observe a strong enrichment of PAR3 at the APM from stage 8 to 10. We can also observe the posterior exclusion of PAR3, at a lower level because of the small size of the posterior domain compared to the anterior domain. The PAR3 APM accumulation can arise in part from the oocyte but also from the nurse cells that transcript PAR3. There is no PAR3 transcription in oocyte, so the contribution of oocyte to the APM accumulation can be only associated with a relocalisation of PAR3 protein. In order to identify the oocyte contribution to the PAR3 anterior accumulation, we performed FRAP experiments on GFP-PAR3 expressed in the oocyte and the nurse cells at stage 9. We suppressed the GFP-PAR3 fluorescence at the APM and in all the nurse cells. Next, we followed the fluorescence recovery to the APM before the recurrence of GFP-PAR3 in the nurse cells due to neosynthesis. We can observe the recovery of the APM signal, which is correlated with a posterior fluorescence decrease. This new result is presented in Figure 8. It suggests that among the fraction of PAR3 accumulated at the APM at least a part of it corresponds to PAR3 already present in oocyte. This process depends on microtubules because if the same experiment is performed in presence of colcemid, the level of recovery is severely affected.

The second point raised by the reviewer concerns the surprising association of PAR3 with membrane. The localisation of PAR3 at the right place does not necessarily require association with membranes. PAR3 could simply diffuse in the cytoplasm. In this work, we do not demonstrate that a membrane association is important to the targeting of PAR3 to the APM. However, we show that i) PAR3 colocalises with PI(4.5)P2 enriched vesicles and that the PI5K SKTL is crucial to the polarised localisation of PAR3, ii) the vesicular trafficking is also important to this localisation and iii) the kinase IKKε which is previously shown implicated in the recycling endosome release of the dynein motor, strongly modifies the PAR3 polarity. Thus, it seems that PAR3 has close links with the vesicular trafficking. We cannot exclude the possibility that PAR3 associates with all PI(4,5)P2 enriched membranes.

Essential revisions:1) The studies of the role of actin and microtubules are rather superficial – for instance the actin data are based only on latrunculin treatment, and the authors assume, based only on the similar phenotype induced by loss of Par1 or expression of a non-phosphorylatable Par3, that the actin network is mediating the effects of Par1 on Par3. This is not a valid conclusion – the same phenotype could be generated through 2 independent mechanisms.

Actin plays a role in the PAR3 distribution control. But we do not know by which mechanism. The actin cytoskeleton controls the polarised localisation of PAR1 (Doerflinger et al., 2006) but also regulate the endocytosis trafficking (Mooren et al., 2012). The two being important for the localisation of PAR3, we cannot conclude on a direct or indirect role of actin on PAR3. This point has been removed from the result section and is addressed only in the discussion.

2) Figure 2J (co-IP of Par3 with DLIC-GFP) is not convincing. The negative control lane input is lower than the positive. Also, a better negative control is needed (another GFP fusion protein); the DLIC-GFP input needs to be shown; and some evidence that the indicated bands actually represent Par3 needs to be provided. And in Figure 2—figure supplement 2B an input control is needed.

As suggested by the reviewer, we have reiterated several times the IP experience of DLICGFP to validate the interaction between DLIC and Par3. We provide a new blot presented in the Figure 8. In the course of those new experiments, as our previous PAR3 antibody made in rat was ran out, we have used a different PAR3 antibody produced in rabbit and kindly provided by A. Wodarz.

Concerning the negative control of this experiment, we have realized two different types of control. As suggested by the reviewer, we have tested the IP with another GFP transgene (PDI-GFP) and have observed no co-IP of PAR3 or DLIC. But we have also performed the co-IP on the same extract with or without anti-GFP antibody. We have chosen to show this last control because if we overexpressed one of these proteins, we can modify the migration profile of others. For example, PAR3 expression modifies the migration profile of DLIC and conversely (variation of postraductional modifications or change in isoforms ratio). In this case, it can be hazardous to compare two wells in which the protein contents are very different. In the case of DLIC/PAR3 IP (Figure 7D), we show now the IP of DLIC-GFP and in control the IP of PDI-GFP.

Although this PAR3 antibody is published, we have further addressed its specificity on PAR3. We have used a *Drosophila* strain that express PAR3-GFP on its own promoter and in absence of endogenous PAR3: BAC PAR3GFP = P[w+, FRT9-2]18E, f, PAR3[815.8], P{CaryP, PB[BAC PAR3- sfGFP2]attP18} (Besson et al., 2015). We have compared the PAR3 protein profile revealed with anti-PAR3 antibody or with anti-GFP antibody. Furthermore, we have also validated the specificity of this antibody by monitoring PAR3 knockdown after RNAi.

This new result is presented in Figure 7—figure supplement 2. As requested by the reviewer we added an input control (using α-tubulin) in Figure 4—figure supplement 1.

3) Figure 3. The authors claim that Par3 particles are associated with PIP2-containing membranes and that Par3 is removed from the PPM by endocytosis on Rab5/PIP2 endosomes, but in Figure 3, most of the Par3 in cytoplasmic spots is NOT associated with Rab5 OR PIP2; and the single Rab5+ vesicle is much larger than any of the PIP2 positive spots. Overall, these data do not agree with the stated hypothesis. The staining for Rab5 and Rab11 is also very diffuse – it would be better to use FP-tagged Rabs for this type of experiment.

We have quantified the amount of PAR3 that colocalises with PIP2 and RAB5 positive particles (by using Manders coefficient / Coloc2 plugin in Fiji). We observed that around 50% of PAR3 co localise with PI(4,5)P2 and 50% co localise with RAB5. We do not know if it is the same 50%. We show in this study that RAB5 is as crucial to the posterior exclusion of PAR3 than the PI5Kinase SKTL. The endocytosis seems important to the exclusion however, we cannot conclude on a direct association of PAR3 with the budding vesicle or on a later association with RAB5 positive vesicles. We have added these news results in the Results section.

The RAB proteins have been tagged with YFP at their endogenous chromosomal loci (Dunst et al., 2015). Like described by Dunst et al., the fluorescence of RAB proteins is very weak in the oocyte and in order to observe it, we must use an anti-GFP antibody to enhance the detection. Unfortunately, this would prevent us from performing a colocalisation with PARGFP.

4) In Figure 3—figure supplement 1, how do we know the antibodies are specific? – the HRS antibody does not seem to detect vesicles.; and the syntaxin16 antibody decorates the plasma membrane but no detectable trans-Golgi.

The antibody anti-HRS and anti-syntaxin16 were now well referenced in the Materials and methods section. The anti-HRS antibody is a gift of H. Bellen and was published in “Hrs Regulates Endosome Membrane Invagination and Tyrosine Kinase Receptor Signaling in *Drosophila”* by (Lloyd et al., 2002). This antibody detects very well the vesicles in follicular cells but there are clearly less vesicles in the oocyte.

The anti-syntaxin16 stained Golgi and lysosome compartments (Akbar et al., 2009). From the stage 9, Golgi begins to accumulate in the subcortical region (Lee and Cooley, 2007). Nevertheless, now we show a magnification on cytoplasm where we can observe some vesicles.

5) The interaction between SKTL and Par3 is based only on over-expressed proteins. The blot needs the GFP-SKTl input and a better (GFP fusion) negative control. Also, I could not find data to support the contention that Par3 forms a tripartite complex with SKTL and PIP2, as is stated in subsection “SKTL and PAR3 act in synergy to form vesicular platforms”. The authors also suggest that Par3 could regulate SKTL activity, and that this would provide a positive feedback loop, but again without any supporting evidence.

We have added the input of the samples, and a control with a GFP alone, rather than a control without anti-GFP.

Concerning the second point, indeed we had no evidence of a tripartite complex. We quantified the colocalisation of PAR3 with PI(4,5)P2 or SKTL (Menders coefficient). Around 50% of PAR3 colocalise with PI(4,5)P2 or SKTL. Since we performed a colocalisation of PAR3, SKTL and PI(4,5)P2 and the result is presented in Figure 3G.

Finally, we have suppressed the positive feedback loop hypothesis of the text.

6) In Figure 5 it looks as though the mutant SKTL is expressed at lower levels than the WT. Could this, rather than loss of kinase activity, explain the lower GFP-Par3 in cytoplasmic spots? The colocalization with Rab5 and 11 is not convincing. Moreover, it looks as though the aAPM has disintegrated – is this not an issue with interpretation of the data?

The Myc-SKTL and Myc-SKTL^DNRQ^ transgenes are inserted at the same site in the genome thanks to phiC31 integrase system (chromosome 3L, position 68A4). The expression of these two transgenes are thus of the same level. We have performed a western blot on ovary extracts and quantified with α-tubulin as loading control.

The results are presented in Author response image 1.

**Author response image 1. respfig1:** Our results show that the mutant form is more stable than wild type. Thus, the absences of effect of DNRQ form cannot be explain by protein instability.

In order to clarify the results concerning the colocalisations, we have quantified the colocalisation of PAR3 with RAB5 and RAB11 vesicles. The Pearson’s coefficient was obtained by using the FiJi Macro Coloc2 on our images. The negative control was provided by quantifying colocalisation for the same images, but after rotation of one by 90 degrees, a condition in which only random colocalisation is observed. The results are now presented in the manuscript Figure 6A. We have also added, in the result part, the quantification of PAR3 that colocalises with RAB5 or RAB11 (Manders ‘s coefficient).

We agree with the reviewer on the difficulties to visualise the APM in Figure 5G. It is just because the plasma membrane is refolded and appears in XY plan. For clarity, this image has been removed with this new version.

7) Finally, I think it is premature to conclude much from the IKK-eta data – clearly knockdown generates some kind of cytoplasmic inclusion in the oocytes, that accumulate actin and Rab11, but it’s not clear what they are; and since Par3-GFP is still strongly accumulated all over the plasma membrane, including the APM (Figure 6A – I), I don't think that it is clear that IKKe is necessary for unloading it from microtubules.

We have reorganised the text concerning IKKe in the result section, we have simplified the conclusion and remove the interpretations. From our results, we can conclude that *ikke* knockdown leads to the accumulation of (PI,4,5)P2 et RAB11 enriched vesicles at the MT minus end and that the PI(4,5)P2 and microtubules are required to this accumulation.

Moreover, PAR distribution is strongly affected by this knockdown.

8) In general, the authors seem to interpret their data as if Par3 is an integral membrane protein that is being endocytosed and transported by the microtubule machinery, but of course it is not – it is a peripheral protein that associates with negatively-charged phospholipids in a highly dynamic way and is presumably able to diffuse through the cytoplasm between vesicles and plasma membrane.

We agree with reviewer and we have modified the Discussion section:

“Although PAR3 does not need a priori an association with membrane to be transported, we found it very frequently close to endocytic membranes. PAR3 on the endosome could be a passenger of the vesicles, which are then transported to the anterior. In parallel, PAR3 could also control its own association with vesicles. Indeed, we hypothesize PAR3 could recruit SKTL that in return produces PI(4,5)P2 that stabilises PAR3 to the membrane and recruits the vesicle budding machinery.” We do not exclude that PAR3 associates with vesicles by simple electrostatic interactions without going by endocytosis. However, the formation of the endocytosis vesicles seems to be important to the final distribution of PAR3.

Reviewer #3:

Jouette and colleagues have examined mechanisms that position the polarity protein Baz/Par-3 in a well-established model of cell polarization, the Drosophila oocyte. They examine a number of mechanisms involved in the cortical removal, sub-cellular trafficking and intracellular compartment release of Baz. The trafficking of Baz through the cytosol is poorly understood, and the results of this paper make significant advances in elucidating this process. However, major concerns should be addressed.Major concerns1) All of the Baz distributions are determined with an over-expressed Baz construct. Thus, it is unclear how relevant identified mechanisms are for endogenous levels of the protein. The newly identified mechanisms should be additionally evaluated with probes for endogenous Baz (using either Baz antibodies or an available Baz-GFP gene trap line).

To identify and to characterize the PAR3 regulation mechanism, firstly we have used a *Drosophila* strain that expresses PAR3-GFP on its own promoter and in absence of endogenous PAR3: BAC PAR3GFP = P[w+, FRT9-2]18E, f, baz[815.8], P{CaryP, PB[BAC Baz- sfGFP2]attP18} (Besson et al., 2015). The level of expression is clearly weaker than with UAS PAR3-GFP expression. We have not used thereafter this strain during this study because of the ubiquitous expression of PAR3. Indeed, the accumulation of PAR3 at the apex of somatic follicular cells around the oocyte masks the potential localisation of PAR3 at the plasma membrane of oocyte, the two membranes being very close. For the same raison, we did not employ PAR3-GFP trap line. Then, the use of antibody is not so easy in oocyte. The follicular cells layer is a natural barrier that prevents the antibody penetration. To stain PAR3, we must permeabilize the membrane with 0,3% Triton X-100 detergent, and the final staining is not enough reproducible to precise quantification.

To clarify this interesting point, we have added images of these experiments in the Figure 1—figure supplement 1A, and precise in the text why we do not continue with these transgenic lines.

In this study, we have used a UAS PAR3-GFP line, expressed with a line in which GAL4 is expressed under the control of maternal α tubulin promoter. Thus, in this condition, PAR3 is expressed only in germline cells and not in follicular cells. We have verified that PAR3GFP expression did not induce phenotype like nucleus mispositioning and we have controlled that the mechanisms described here was always saturable by a PAR3 expression increase. We agree with the reviewer that our conditions are not the best, but it is the one that we can use to answer to this precise question.

2) Connections between the mechanisms that remove Baz from the cortex are unclear. Possible connections should be investigated with the tools already used to determine if the mechanisms are independent or interdependent.2.1) Abnormal Baz accumulations arise along the later cortex with dynein RNAi. Are these accumulations dependent on cortical removal steps mediated by Par-1, PIP2 or Rab5? Perturbations of these proteins in pair-wise combinations could reveal dependent or independent cortical removal mechanisms.

As suggested by the reviewer, we tried to combine the perturbations induced by these proteins on PAR3 localisation despite the genetic association difficulties.

First of all, to test whether PAR3 accumulation along the lateral cortex observed in *dhc64c* knockdown was dependent on PAR1, we expressed PAR3-GFP with RNAi PAR1 and RNAi DHC64c. The combined expression of these transgenes strongly affects the morphology and the development of the egg chamber. As it can be observed (Author response image 2), the oocyte in the egg chamber present an abnormal small size compared to the nurse cells. Moreover, this phenotype is also observed and seems stronger, when PAR3AA (a form non phosphorylable by PAR1) is overexpressed in *dhc64c* knock-down context. These kinds of phenotypes are usually related to a defect in the growth of the oocyte due to a transport failure from the nurse cells to the oocyte. Unfortunately, these interactions make it difficult to quantify PAR3 polarity during oogenesis and it is therefore difficult to conclude about the interdependence between these mechanisms.

**Author response image 2. respfig2:** 

We also tried to test the interdependence between DHC64c and SKTL, or RAB5. Sadly, the expression of RNAi DHC64c with either RAB5 DN (Dominant Negative) or in context *sktl* mutant, disorganise the structure of the egg chamber making the quantification impossible. To test the involvement of endocytosis in these processes, we also used the drug dynasore, which is an inhibitor of dynamin and thus of the endocytosis vesicle scission. However, this drug is associated with pleiotropic phenotypes in the oocyte and we were not able to answer this point to the reviewer.

2.2) Skittles overexpression is shown to increase the removal of Baz from the posterior domain. Does this removal require Par-1 or Rab-5? Again, combining the Skittles overexpression with the Par-1 RNAi or the Rab5DN could address how interdependent the removal mechanisms are.

We also tried to address the interdependence between SKTL and PAR1 or RAB5 for PAR3 posterior exclusion.

Firstly, we combined the expression of SKTL (UAS-myc-SKTL) with PAR1 knockdown (UAS RNAi PAR1) and we observed a genetic interaction between them concerning PAR3 posterior exclusion. We showed that SKTL overexpression is always able to exclude PAR3 of the posterior in PAR1 RNAi knockdown. The same observation was realised by using PAR3 AA mutant form (non phosphorylable by PAR1). This new data is presented Figure 4.

To observe the interdependence between SKTL and RAB5, we expressed PAR3-GFP with Myc-SKTL and RAB5DN (Dominant Negative). However, as illustrated in the image below, the expression of all these proteins induces precocious oogenesis defects. The PAR3 quantification was therefore not possible.

**Author response image 3. respfig3:** 

3) Since Baz becomes isotropic around the cortex with IKK-epsilon knock down, does IKK-epsilon also play a role in the removal of Baz from the posterior cortex? If IKK-epsilon only promoted delivery to the anterior cortex, then removal from the posterior would still be expected. Some assessment of IKK-epsilon's direct or indirect role at the posterior would be expected within the overall context of the paper.

The kinase IKKε has a role probably more complex that the one described in this work. It seems for example, to have a slight effect on PAR1 posterior localisation. Therefore, we have chosen to focus on the already described phenotype of IKKε, the ectopic accumulation of RAB11 cargo at the microtubule minus ends (Otani et al., 2011).

[Editors' note: the author responses to the re-review follow.]

This manuscript (an earlier version of which which has been previously reviewed by eLife but is now being treated as a new submission rather than a resubmission) provides evidence for a pathway that polarizes the distribution of Baz (Par-3) across the Drosophila oocyte, an important model of cell polarity. The model is an interesting and, if fully supported by the data, would be appropriate for publication in eLife. However, there are significant concerns that would need to be addressed before it could be accepted.Essential revisions:1) Generally, the quality of the biochemical evidence is not as strong as it should be. For example, in Figure 7E and F, the IPs of GFP-SKTL testing for pull-down of PAR3 and DLIC, the control was no addition of GFP antibody. Since addition of antibodies can increase non-specific binding, the better control is to IP an irrelevant GFP protein with the GFP antibody (as done in Figure 7D). Especially when the protein interaction they are trying to show is weak, good controls are required.

The immunoprecipitation experiment was reiterated with another GFP-protein in control (GFP-SNAP). The GFP-SNAP protein does not precipitate PAR3 or DLIC. The new IP blot is presented in Figure 7E and the charge control has been added in Figure 7—figure supplement 2.

Even in 7D, which does have an appropriate control, the co-IP is pretty weak. This does not invalidate it, but it should be acknowledged in the text.

This precision has been added in subsection “DLIC, PAR3 and SKTL are interacting partners”.

In the earlier version it was unclear that the 250kD band that was observed on western blots actually corresponded to Par3. The authors have used RNAi strains to address this point and the conclusion is a bit more convincing. It is still hard to fully appreciate, though. For instance, in Figure 7—figure supplement 2, a strain over-expressing Par3-GFP using the UAS-Gal4 driver is used. These embryos should be expressing a relatively high level of Par-GFP. In the total fraction from these embryos the most prominent band is around 100kD when probed with a Par3 antibody. However, this band is hard to detect with the GFP antibody. In fact, it is hard to see any bands in the GFP blot. Next, the sample was immunoprecipitated. It is unclear from the legend which antibody (Par3 or GFP) was used in the IP. In the immunoprecipitated fractions, the most prominent band is the 250kD band. The band is reduced in the strain expressing Par3 RNAi but it is not clear how it can be detected so prominently in the IP fraction when the same isoform is a minor species at best in the total fraction. Is the IP antibody somehow specifically precipitating an oligomeric form? The experiment with the endogenously expressed BAC-based Par3-GFP construct is more convincing.2) Another major issue is the over-reliance on overexpressed Par3. Key results should be repeated to look at endogenous Par3 expression. For example, are the Par3-positive cytoplasmic particles, and effects on these particles, observed when Par3 is expressed by its own regulatory regions? Without this additional data, there is a risk that the effects they report are artifacts of overexpression.

To address this issue, we have used a *Drosophila* strain, PAR3-Trap, in which a GFP is inserted in frame in PAR3 (Januschke and Gonzales, 2010). We have added in a new figure (Figure 2—figure supplement 2) the illustration of PAR3 trap line that shows the presence of dotted structure in the oocyte cytoplasm. This observation was realized on previtellogenic stage because the accumulation of autofluorescent vitellus masks the presence of these structures thereafter. Furthermore, we have assayed the microtubule requirement for the cytoplasmic transport of these structures. We performed a colchicin treatment on PAR3 trap egg chambers. We then observed that PAR3 accumulates in numerous dotted structures in the cytoplasm, as observed previously with the PAR3-GFP overexpression in presence of colchicin. The expression of PAR3GFP in the follicular cells with this strain precludes the analysis of PAR3-GFP at the oocyte plasma membrane. These new results are presented in the Figure 2—figure supplement 2.

3) The results using a dominant negative rab5 should be confirmed with an shRNA construct (available from Bloomington). There is a risk that the DN might produce off target effects.

We have quantified PAR3 distribution in a RAB5 RNAi context. Among the twodrosophila transgenic lines available in Bloomington stock center, the BL34832 stock affects most strongly endocytosis during oogenesis according defect in yolk uptake (Compagnon et al., 2009). RAB5 RNAi alters PAR3 distribution in the same manner that RAB5 dominant negative but a little less strongly. This new result is present in Figure 5.